# Cascaded dissipative DNAzyme-driven layered networks guide transient replication of coded-strands as gene models

Jianbang Wang[1,2], Zhenzhen Li[1,2] & Itamar Willner [1] ✉

Dynamic, transient, out-of-equilibrium networks guide cellular genetic, metabolic or signaling processes. Designing synthetic networks emulating natural processes imposes important challenges including the ordered connectivity of transient reaction modules, engineering of the appropriate balance between production and depletion of reaction constituents, and coupling of the reaction modules with emerging chemical functions dictated by the networks. Here we introduce the assembly of three coupled reaction modules executing a cascaded dynamic process leading to the transient formation and depletion of three different $Mg^{2+}$-ion-dependent DNAzymes. The transient operation of the DNAzyme in one layer triggers the dynamic activation of the DNAzyme in the subsequent layer, leading to a three-layer transient catalytic cascade. The kinetics of the transient cascade is computationally simulated. The cascaded network is coupled to a polymerization/nicking DNA machinery guiding transient synthesis of three coded strands acting as "gene models", and to the rolling circle polymerization machinery leading to the transient synthesis of fluorescent Zn(II)-PPIX/G-quadruplex chains or hemin/G-quadruplex catalytic wires.

Reactive networks play important roles in controlling the dynamics of living systems by guiding cellular genetic, metabolic or signaling processes[1,2]. Particularly, genetic reaction networks comprising of interconnected genes responding to auxiliary stimuli process complex cellular functional modules revealing adaptive[3], pulsed[4], oscillatory[5–7], temporal[8] and multi-stable behaviors[9] leading to dictated gene expression[10], cell differentiation[11], morphogenesis[12] or eventually to genome instability and oncogene-induced replication stress and cancer[13–16]. In vitro rational design of chemical circuits emulating native reaction network attracts growing interest within the broad field of Systems Chemistry[17–20]. The information encoded in the base sequence of nucleic acids leads to design modularity of ingenious nucleic acid structures[21], unique recognition and catalytic activities of sequence-tailored strands, e.g., aptamers[22,23], DNAzymes[24,25] or nucleoapzymes[26], and guided responsiveness of DNA strands to enzymes or auxiliary triggers[27,28], such as strand displacement[29,30], pH[31,32], ions[33–35] or light[36],

that allow the reconfiguration of DNA structures. These features of nucleic acids enable the construction of DNA-based assemblies and networks duplicating functions of native networks. Indeed, different DNA-based networks revealing adaptive[37], oscillatory[6,38] and bistability[39] properties were reported. Emerging applications of adaptive and dynamic networks were realized, including the dynamic self-assembly of DNA nanostructures[40] or the spatiotemporal triggered compartmentalization of gene circuits[41]. Dynamic networks mimicking complex signal-triggered native networks were demonstrated[42]. These included the development of constitutional dynamic networks revealing adaptive[43,44], hierarchical adaptive[45], feedback-driven[46] and intercommunicating networks[47]. Different applications of constitutional dynamic networks were addressed, such as network-guided activation of biocatalytic cascades[48] and dynamic network dictating material properties, such as controlled stiffness of hydrogels for switchable and guided drug release[49]. In addition, an

---

[1]The Institute of Chemistry, The Center for Nanoscience and Nanotechnology, The Hebrew University of Jerusalem, 91904 Jerusalem, Israel. [2]These authors contributed equally: Jianbang Wang, Zhenzhen Li. ✉e-mail: Itamar.Willner@mail.huji.ac.il

important class of dynamic networks, emulating biological networks, includes DNA-based out-of-equilibrium, dissipative, signal-triggered transient systems. These included the design of transcriptional oscillators[50] or switches, bistable regulatory networks[51], and dynamic DNA networks that are coupled to enzymes, e.g., polymerase/endonucleases/nickase, leading to oscillatory behavior[6,52] or to transient reconfiguration of constitutional dynamic network and secondary transient network-controlled catalysis[53]. Also, fuel-triggered aptamer-based and enzyme-guided release and uptake of loads[54], photo acid-driven dissipative polymerization/depolymerization of DNA fibers[55] and light controlled out-of-equilibrium DNA ligation cycles[56] were demonstrated. Different applications of fuel-triggered transient enzyme-coupled (nickase) nucleic acid-based modules were reported, including the design of gated transient networks[57] and the triggered transient operation of photoinduced electron transfer cascades[58].

Although the in vitro rational design of chemical circuits mimicking natural reaction networks was achieved, the future design of synthetic networks and particularly, transient, dissipative, networks, imposes important challenges including the ordered connectivity of reaction modules, the engineering of the appropriate balance between the production/depletion of dynamic reaction species and specifically, the coupling between dynamic networks and emerging functions dictated by the networks.

In the present study, we report on the design of a dynamic three-layer DNAzyme cascade. We present the organization of three coupled reaction network modules where the triggered activation of the first module activates the generation of a transient operating DNAzyme that initiates the fueled operation of the subsequent transient cascade composed of layered DNAzymes networks. Furthermore, we demonstrate the use of the transient biocatalytic cascade as a functional unit to drive transient catalytic processes, such as the rolling circle polymerization of DNAzyme wires or the network-triggered transient replication of target nucleic acid strands as a model for the replication of genes in nature. Furthermore, we note that throughout the report, we consider each of the dissipative DNAzyme layer as a supramolecular reaction module that is activated by the supply of energy in the form of a fuel strand that generates waste products and a non-equilibrated reaction intermediate that undergoes transient transition to the initial self-assembled reaction module. The kinetic features of the layered DNAzyme cascade are modeled by computational simulations. The kinetic analysis of the systems enables to predict the behavior of the transient processes under different auxiliary conditions. It should be noted that in contrast to previous reports that demonstrated enzyme-controlled transient nucleic acid cascades, we introduce all-DNAzyme-driven layered dissipative networks. Beyond the enhanced complexity introduced by these systems, we emphasizes the "coupler" element as a fundamental concept to control the dynamic communication between the layers. In addition, the coupling of the transient network cascade to subsequent synthesis of artificial functional coded-strands model gene replication machineries provides a step towards emerging functionalities dictated by transient networks.

## Results

### Design and operation of transient DNAzyme systems

Figure 1a outlines the composition and mode of operation of the first layer of the transient DNAzyme-triggered process. The reaction module I consists of the toehold-functionalized duplex A/A', the hairpin $H_1$ and $Mg^{2+}$ ions. Strand A is modified at its 5' end with the fluorophore Cy5 and strands A' at its 3' end with the quencher BHQ2. Subjecting the reaction module I to the trigger $T_1$ results in the formation of $T_1$/A duplex by displacing the quencher-modified strand A' from the constituent AA', leading to the switched "ON" fluorescence of the duplex $T_1$/A. The resulting $T_1$/A duplex includes, however, in the free toehold tethers the base sequences that correspond to the $Mg^{2+}$-ion-dependent

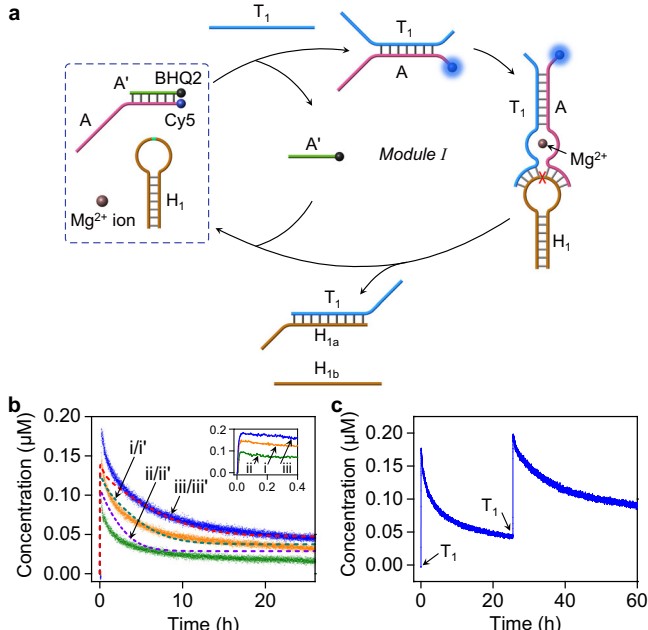

**Fig. 1 | Triggered transient formation and depletion of a DNAzyme. a** Triggered transient operation of a reaction module consisting of duplex A/A' (0.2 μM), hairpin $H_1$ (1 μM) and $Mg^{2+}$-ions, resulting in a $Mg^{2+}$-ion-dependent DNAzyme $T_1$/A. **b** Transient concentrations corresponding to: (i)-solid line, the formation and depletion of the $Mg^{2+}$-ion-dependent DNAzyme $T_1$/A in the presence of trigger $T_1 = 0.3$ μM, and (i')-dashed line, computationally-simulated transient using the kinetic model presented in Supplementary Fig. 2. Transient and computational predicted curves (ii)/(ii'), corresponding to the $T_1$-triggered formation/depletion of DNAzyme $T_1$/A in the presence of $T_1 = 0.2$ μM, and (iii)/(iii') transient and computationally predicted curves, corresponding to the formation/depletion of DNAzyme $T_1$/A in the presence of $T_1 = 0.4$ μM. Insert: Time dependent dynamic evolution of $T_1$/A at short time intervals: (i) in the presence of trigger $T_1 = 0.3$ μM, (ii) in the presence of trigger $T_1 = 0.2$ μM, (iii) in the presence of trigger $T_1 = 0.4$ μM (For the half-life values of the transients, see Supplementary Fig. 6.). **c** Cyclic operation of the transient DNAzyme cycle ($T_1 = 0.4$ μM) shown in **a**. Arrows indicate the addition of the trigger $T_1$. Source data are provided as a Source Data file.

DNAzyme that binds the hairpin $H_1$ acting as substrate of the DNA-zyme. Cleavage of the hairpin yields two fragments $H_{1a}$ and $H_{1b}$. The cleaved $H_{1a}$ displaces $T_1$ from the complex $T_1$/A to form $T_1$/$H_{1a}$, and the released A rebinds to A' recovering the original duplex A/A' of the module I, where the fluorescence of Cy5 is quenched by BHQ2. Thus, subjecting the reaction module I to the fuel strand $T_1$ leads to the transient formation of the catalytically active supramolecular DNA-zyme structure $T_1$/A that cleaves the hairpin substrate to recover the original module by the depletion of the DNAzyme and the concomitant formation of $T_1$/$H_{1a}$ and $H_{1b}$ as the waste products. The transient formation and depletion of the catalytic DNAzyme structure are followed by the transient fluorescence features of Cy5. Using an appropriate calibration curve relating the fluorescence intensities of the Cy5 constituent to its concentrations (Supplementary Fig. 1), the transient fluorescence changes of the system are translated into concentration changes corresponding to the transient formation and depletion of the complex $T_1$/A, Fig. 1b, curve (i). A kinetic model that accounts for the formation and depletion of the catalytically active DNAzyme was formulated, Supplementary Fig. 2. The experimental transient curve (i) shown in Fig. 1b was computationally simulated using the kinetic model, and the fitted computational curve (i') is overlaid on the experimental curve. The set of computationally simulated rate constants formulated by the kinetic model are tabulated in Supplementary Table 1. The kinetic features of the transient system shown in Fig. 1a should depend on the concentrations of the triggering fuel $T_1$.

Accordingly, the set of rate constants tabulated in Supplementary Table 1 was used to predict the transient behavior of $T_1/A$ at trigger concentrations corresponding to 0.2 μM and 0.4 μM and the simulated results are presented in curves (ii') and (iii'). The predicted results were experimentally validated, curves (ii) and (iii), respectively. As the concentration of $T_1$ increases, the peak content of the transient complex $T_1/A$ is higher, and the recovery rate of the rest module is slower. Very good fit between the predicted and experimental results is observed, demonstrating the value of the predictive model to follow the transient behavior of the system (for further support of the simulated kinetic model by electrophoretic experiments, see Supplementary Fig. 3 and accompanying discussion). The transient features of the module I can be recycled by re-addition of the fuel strand $T_1$, Fig. 1c marked with an arrow. For experimental validation of some of the partial rate constants derived by the computational simulations, and further control experiments supporting the transient mechanism of DNAzyme $T_1/A$, see Supplementary Figs. 4, 5 and accompanying discussions. We note that the recovery of the parent module by the transient process is slightly incomplete. This is attributed to the hybridization of the waste product $H_{1b}$ with $H_{1a}$ a process inhibiting the full recovery of the initial state.

Following the concept introduced to develop the transient formation of the DNAzyme $T_1/A$, two other transient DNAzyme modules, II and III were designed (Supplementary Figs. 7 and 12). Module II consists of the duplex B/B', where strands B and B' are modified with FAM fluorophore and BHQ1 quencher, respectively, and the hairpin $H_2$ is integrated in the system. Subjecting module II to trigger $T_2$ activates the transient formation and depletion of DNAzyme $T_2/B$. The transient operation of DNAzyme $T_2/B$ is presented in Supplementary Fig. 7b together with the computational simulations of the dynamic processes using the kinetic model, Supplementary Fig. 9. The calibration curve allowing the translation of the fluorescence changes of $T_2/B$ into transient concentration changes, the kinetic model to simulate module II, the derived rate constants, and further support of the kinetic model by experiments are summarized in Supplementary Figs. 8–11 and Supplementary Table 2, respectively. The computational simulations were used to predict and experimentally validate the operation of the system at different auxiliary conditions, and the cyclic transient operation of DNAzyme $T_2/B$ (Supplementary Fig. 7c) and the half-life values of the transients (Supplementary Fig. 7d, e) were demonstrated. Similarly, module III consisting of the duplex C/C' modified with Cy3 and BHQ2 and hairpin $H_3$ was triggered by $T_3$ to yield the transient formation and depletion of DNAzyme $T_3/C$ (Supplementary Fig. 12). The transient dissipative operation of DNAzyme $T_3/C$, and the computational simulation of the dynamic system using the kinetic model (Supplementary Fig. 14) are presented in Supplementary Fig. 12b. For the complementary experiments describing the kinetic model for computational simulation of module III, the derived rate constants of the transient operation of module III, see Supplementary Figs. 13–15 and Supplementary Table 3. As before, the computational simulations were used to predict and experimentally validate the operation of the system at different auxiliary concentrations. Furthermore, the cyclic transient operation of DNAzyme $T_3/C$ (Supplementary Fig. 12c), the half-life values of the transients (Supplementary Fig. 12d, e) and supplementary control experiment following DNAzyme $T_3/C$ and accompanying discussion (Supplementary Fig. 16) are depicted.

### Operation of bilayer-transient DNAzyme cascades

The successful operation of separate transient DNAzymes was then applied to organize, in the first step, a bilayer transient DNAzyme cascades, and subsequently a three-layer transient DNAzyme cascade. Figure 2a depicts the cascading of transient modules I and II. To operate the transient two DNAzymes cascade, the duplex $S/T_2$ is introduced, as coupler unit that interconnects modules I and II. The

cascade is triggered by $T_1$ that activates module I and yields the DNAzyme $T_1/A$. Cleavage of hairpin $H_1$ yields the fragmented products $H_{1a}$ and $H_{1b}$, where $H_{1a}$ is used to operate the transient depletion of $T_1/A$, and the fragmented product $H_{1b}$ displaces duplex $S/T_2$ to yield $H_{1b}/S$ and to release the trigger $T_2$. Trigger $T_2$ activates module II leading to the transient operation of DNAzyme $T_2/B$. It should be noted that the coupling unit $S/T_2$ is essential to communicate the two layers. Direct coupling of $H_{1b}$ with the second layer would require complementarity between $H_{1b}$ and B and, thus, an immediate cross-talk between B and $H_1$ which would perturb the communication process. The two-layer DNAzyme cascade is followed by the transient fluorescence of Cy5 and FAM associated with the two dissipative DNAzymes $T_1/A$ and $T_2/B$, respectively. Figure 2b shows the transient concentration changes corresponding to the formation and depletion of cascaded transient DNAzymes $T_1/A$ and $T_2/B$ upon triggering the system with $T_1$ ($T_1 = 1.2$ μM) (Supplementary Fig. 17). The transient formation of the DNAzyme $T_1/A$, Fig. 2b, curve (i), shows a rapid built-up of $T_1/A$ that reaches a peak value after ca. 13 min followed by a decay proceeding for a substantially longer time-interval leading to the rest state of module I. On the other hand, the built-up of the second DNAzyme $T_2/B$ proceeds for a substantially longer time-interval and reaches a peak value only after 156 min, Fig. 2b, curve (ii), and afterwards, shows a slow decay regenerating the original module II (see also Fig. 2b, inset). Control experiments revealed that upon exclusion of the coupler unit, $S/T_2$, only module I was activated, whereas module II was switched off (Supplementary Fig. 19). In addition, subjecting module II to the coupler $S/T_2$ does not affect module II, Supplementary Fig. 20, implying that $S/T_2$ reveals the duplex stability that eliminates the release of traces of $T_2$ to activate module II. Also, subjecting module II alone to trigger $T_1$ did not activate module II, Supplementary Fig. 21. Besides, addition of hairpin $H_1$ to module II does not activate layer 2, Supplementary Fig. 22. The kinetic models corresponding to the individual transient DNAzymes $T_1/A$ and $T_2/B$ were integrated to include the $S/T_2$ coupling event and the kinetic model corresponding to the cascade of modules I and II is presented in Supplementary Fig. 18. The model was adapted to computationally simulate the transients corresponding to the cascaded DNAzymes $T_1/A$ and $T_2/B$, curves (i') and (ii'), Fig. 2b. The derived rate constants are summarized in Supplementary Table 4. The resulting rate constants were applied to predict the transient dissipative behavior of DNAzymes $T_1/A$ and $T_2/B$ upon subjecting the cascaded modules I and II to a $T_1$ fuel-trigger concentration of 1.8 μM, Fig. 2d, curves (i') and (ii'), respectively, and the predictive transients were validated experimentally, curves (i) and (ii), respectively. Very good fit between the predicted transients and the experimental curves is demonstrated. Note that increasing the concentration of the trigger $T_1$ intensifies, the peak content of $T_2/B$ and shortens the time for generating the peak content to 108 min (as compared to the lower concentration of $T_1$ that required 156 min to yield the peak content of $T_2/B$), see Fig. 2d, inset. In addition, by successive addition of $T_1$, the cyclic operation of the bilayer DNAzyme cascade was demonstrated, Fig. 2f. Knowing the initial concentrations of module I and II, and the transient concentrations of the DNAzyme constituents, the respective transient yields of the constituents were evaluated. Accordingly, Fig. 2c, e, g present the transient yields of the DNAzyme constituents upon operating the cascade.

Similarly, the two-layer cascade consisting of modules II and III, using the coupler $M/T_3$ was examined (Supplementary Fig. 24). In this system, trigger $T_2$ activates the transient operation of DNAzyme $T_2/B$ where cleavage of $H_2$ results in $H_{2a}$ that recovers module II and yields $H_{2b}$ that displaces the coupler $M/T_3$ and releases trigger $T_3$ that activates the transient operation of DNAzyme $T_3/C$. Cleavage of $H_3$ recovers, then, module III. The transient cascaded operation and computationally simulated concentration changes of DNAzymes $T_2/B$

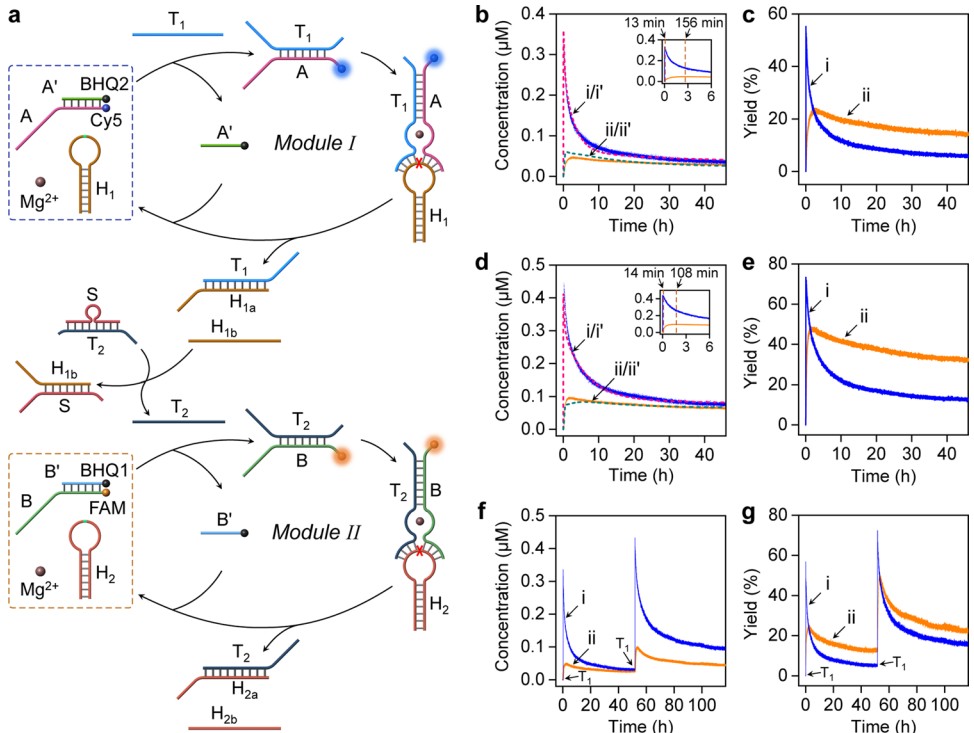

**Fig. 2 | Triggered operation of a two-layer transient DNAzyme cascade.**
**a** Schematic $T_1$-triggered operation of the cascaded module I and II using S/$T_2$ as coupler. The cascade involves the simultaneous operation of transient DNAzymes $T_1$/A and $T_2$/B. **b** Transient concentration changes in the presence of $T_1$ = 1.2 μM (A/A′ 0.6 μM, $H_1$ 3 μM, B/B′ 0.2 μM, $H_2$ 1 μM, S/$T_2$ 1.2 μM) corresponding to: DNAzyme $T_1$/A - (i-solid line) experimental result; (i′-dashed line) computationally simulated result. DNAzyme $T_2$/B - (ii-solid line) experimental result; (ii′-dashed line) computationally simulated result. Inset: Transient curves at short time-intervals of the transient curves. **c** Transient yields of: (i) DNAzyme $T_1$/A; (ii) DNAzyme $T_2$/B. Results were recorded in the presence of $T_1$ = 1.2 μM (A/A′ 0.6 μM, $H_1$ 3 μM, B/B′ 0.2 μM, $H_2$ 1 μM, S/$T_2$ 1.2 μM). **d** Transient concentration changes of the cascaded DNAzyme in

the presence of $T_1$ = 1.8 μM (A/A′ 0.6 μM, $H_1$ 3 μM, B/B′ 0.2 μM, $H_2$ 1 μM, S/$T_2$ 1.8 μM): DNAzyme $T_1$/A-(i′-dashed line) computationally predicted transient, (i-solid line) experimental result. DNAzyme $T_2$/B-(ii′-dashed line) and (ii-solid line) computationally predicted and experimental transients, respectively. Inset-Transient short time scales. **e** Transient yields of: (i) DNAzyme $T_1$/A, (ii) DNAzyme $T_2$/B. Results were recorded in the presence of $T_1$ = 1.8 μM (A/A′ 0.6 μM, $H_1$ 3 μM, B/B′ 0.2 μM, $H_2$ 1 μM, S/$T_2$ 1.8 μM). Cyclic transient concentration changes (**f**) and transient yield (**g**) of: (i) DNAzyme $T_1$/A; (ii) DNAzyme $T_2$/B by re-adding the trigger $T_1$ = 1.2 μM. (For half-life of the transient DNAzyme $T_1$/A, see Supplementary Fig. 23.) Source data are provided as a Source Data file.

and $T_3$/C (Supplementary Figs. 26, 27) are displayed in Supplementary Fig. 24b, c. The set of computationally simulated rate constants formulated by the kinetic model are tabulated in Supplementary Table 5. The cyclic operation of the $T_2$- triggered transient dynamic cascaded modules II and III and the half-life of the transient DNAzyme $T_2$/B are introduced in Supplementary Figs. 24d, 25, respectively. Complementary control experiments supporting the cascaded dynamic process of module II and III are shown in Supplementary Figs. 28, 29, where exclusion of the coupler M/$T_3$ prevents the communication between module II and module III, and $T_2$ alone does not affect module III. Also, subjecting module III to the coupler M/$T_3$ does not affect module III, Supplementary Fig. 30, implying that M/$T_3$ reveals the duplex stability that eliminates the release of traces of $T_3$ to activate module III. In addition, subjecting module III to $H_2$ does not activate module III, Supplementary Fig. 31. It should be noted that the single layer and double layer dissipative reaction moduli can be recycled by consecutive additions of the triggering fuels. Nonetheless as the number of consecutive cycles increases, the dissipative recovery patterns and the reaction moduli are perturbed to the consumption of the hairpins and presumably to the accumulation of the waste strands, e.g. $H_{1b}$, $H_{2b}$, $H_{3b}$ that recombine with the reaction intermediates.

### Operation of a three-layer transient DNAzyme cascade
The final step has involved the operation of the three-layer cascade, Fig. 3a. In this system, the three-layer modules I, II and III were fueled by trigger $T_1$, and the operation of the dynamic dissipative cascade was examined in the presence of the two coupler units S/$T_2$ and M/$T_3$. The

$T_1$ fueled activation of the module I yields the DNAzyme $T_1$/A that cleaves hairpin $H_1$ leading to the transient operation of the first layer and the dynamic activation of module II. The DNAzyme $T_1$/A guided cleavage of hairpin $H_1$ yields $H_{1a}$ and $H_{1b}$, where $H_{1a}$ operates the transient module I and $H_{1b}$ induces displacement of the coupler S/$T_2$ activating the transient operation of the second layer consisting of module II. The transient operation of DNAzyme $T_2$/B results in the cleavage of hairpin $H_2$, where the cleaved product $H_{2a}$ guides the transient operation of module II and the fragmented $H_{2b}$ displaces coupler M/$T_3$ that activates the third layer, module III, where DNAzyme $T_3$/C cleaves hairpin $H_3$ yielding fragment $H_{3a}$ that operates the transient third layer-module III, and the fragment $H_{3b}$ that displaces duplex W/P, yielding a free strand P for further desired usage, vibe infra. The transient operation of the three-layer cascade is followed by the transient fluorescence changes of the free fluorophores associated with the three layers and these were translated to transient concentration changes of the DNAzymes, using appropriate calibration curves (Supplementary Figs. 32, 26 and 13). Figure 3b–d displays the transient concentration changes of the three DNAzymes upon the $T_1$-triggered operation of the three-layer cascade ($T_1$ = 1.8 μM), where curves (i), (ii), (iii)-solid lines correspond to the experimental transients and (i′), (ii′), (iii′) correspond to the computationally simulated results, using the kinetic model formulated in Supplementary Fig. 33. Figure 3e presents the overlay of the transient concentration changes of the DNAzymes consisting the three-layer cascade. The kinetic model and the computationally simulated rate constants (Supplementary Table 6) were then applied to predict the dynamic behavior of the

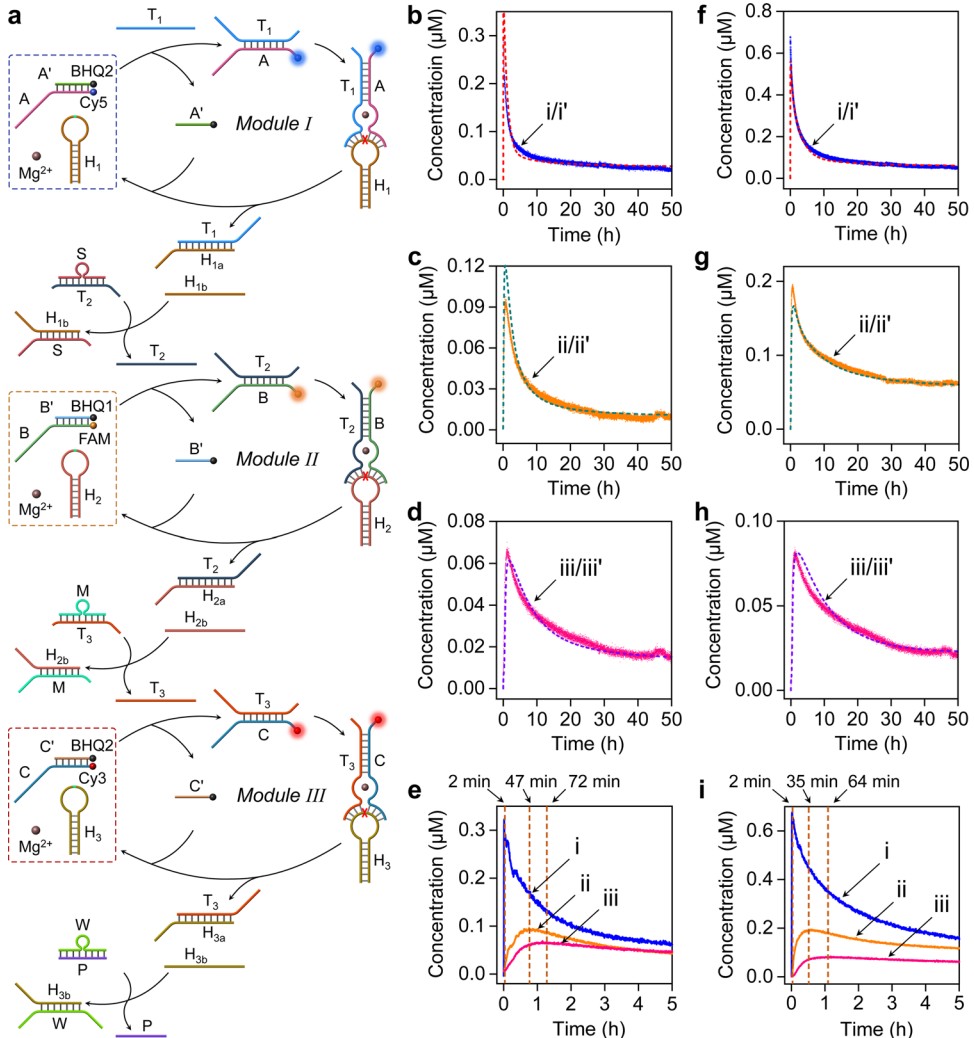

**Fig. 3 | Triggered operation of a three-layer transient DNAzyme cascade.**
**a** Schematic $T_1$-triggered transient operation of the three-layer DNAzyme cascade. The dynamic, transient formation and depletion of DNAzyme $T_1$/A, $T_2$/B and $T_3$/C are followed by the transient fluorescence changes of the fluorophores associated with the dynamic modules. **b–e** Time-dependent transient concentration changes of the DNAzyme constituents in the three-layer cascade triggered by $T_1 = 1.8\,\mu M$ (A/ A' 1.8 μM, $H_1$ 9 μM, S/$T_2$ 1.8 μM, B/B' 0.6 μM, $H_2$ 3 μM, M/$T_3$ 0.6 μM, C/C' 0.2 μM, $H_3$ 1 μM, W/P 2 μM): (**b**)-DNAzyme $T_1$/A operating in module I: Experimental transient (i)-solid curve; computational simulated transient using the kinetic model, Supplementary Fig. 33, (i')-dashed line. (**c**)-DNAzyme $T_2$/B operating in module II, (ii)- experimental transient, (ii')-computational simulated transient. (**d**)-DNAzyme $T_3$/C operating in module III, (iii)-experimental transient, (iii')-computational simulated

transient. (**e**)-Overlay of the transient, dissipative, concentrations of the three DNAzymes participating in the cascade. **f–i**, Transient concentration changes of the DNAzymes in the cascaded system predicted by the kinetic model, Supplementary Fig. 33 and the derived rate constants, Supplementary Table 6, upon triggering the system, $T_1 = 3.6\,\mu M$ (A/A' 1.8 μM, $H_1$ 9 μM, S/$T_2$ 3.6 μM, B/B' 0.6 μM, $H_2$ 3 μM, M/$T_3$ 1.2 μM, C/C' 0.2 μM, $H_3$ 1 μM, W/P 2 μM), dashed curves and experimentally vali- dated transients, solid curves: (**f**), (i)/(i')- DNAzyme $T_1$/A operating in module I; (**g**), (ii)/(ii')- DNAzyme $T_2$/B operating in module II; (**h**), (iii)/(iii')- DNAzyme $T_3$/C oper- ating in module III. (**i**)-Overlay of the dynamic concentration changes of (i) DNA- zyme $T_1$/A, (ii) DNAzyme $T_2$/B and (iii) DNAzyme $T_3$/C, upon subjecting the three- layer modules to $T_1 = 3.6\,\mu M$. (For half-life of the three-layer cascaded DNAzymes, see Supplementary Fig. 35.) Source data are provided as a Source Data file.

three-layer cascade at different auxiliary conditions. Figure 3f–h depicts the predicted concentration changes of the three DNAzymes upon the $T_1$-triggered activation of the cascade ($T_1 = 3.6\,\mu M$), curves (i'), (ii') and (iii')-dashed lines, and the experimentally validated results, curves (i), (ii), (iii). Very good agreement between the dynamic experimental curves and the predicted curves exists. Figure 3i shows the overlay of the experimental transient of the three DNAzymes upon triggering the cascade at $T_1 = 3.6\,\mu M$. Increasing the concentration of $T_1$ from 1.8 μM to 3.6 μM doubles the peak contents of the inter- mediate DNAzymes generated in the system (for the time differences in reaching the peak transient concentrations of the two-layer cascade vs. the three-layer cascade, see Supplementary Discussion, page 43). For further control experiments associated with the stepwise opera- tion of the three-layer cascade see Supplementary Fig. 34 and accompanying discussion.

## Three-layer DNAzyme cascade-guided transient DNA machineries

The study addressed till now the dynamic transient dissipative operation of the cascaded layered network driven by three DNAzymes. To emulate, however, biological systems, where transient cascaded networks control biological functions, we searched for systems that couple the dynamic cascaded network to auxiliary dissipative func- tionalities. Bioinspired by nature, where dynamic time-dependent networks dictates gene replication, we coupled the transient three- layer cascaded network to the dynamically controlled formation of three different, artificially-coded strands mimicking three different genes, Fig. 4a. The strand C' generated as a transient constituent of the three-layer cascade, was used as promoter to activate three transient machineries coding for different functional target strands. Three scaffolds $J_1$, $J_2$, and $J_3$ that include common recognition sites for C' were

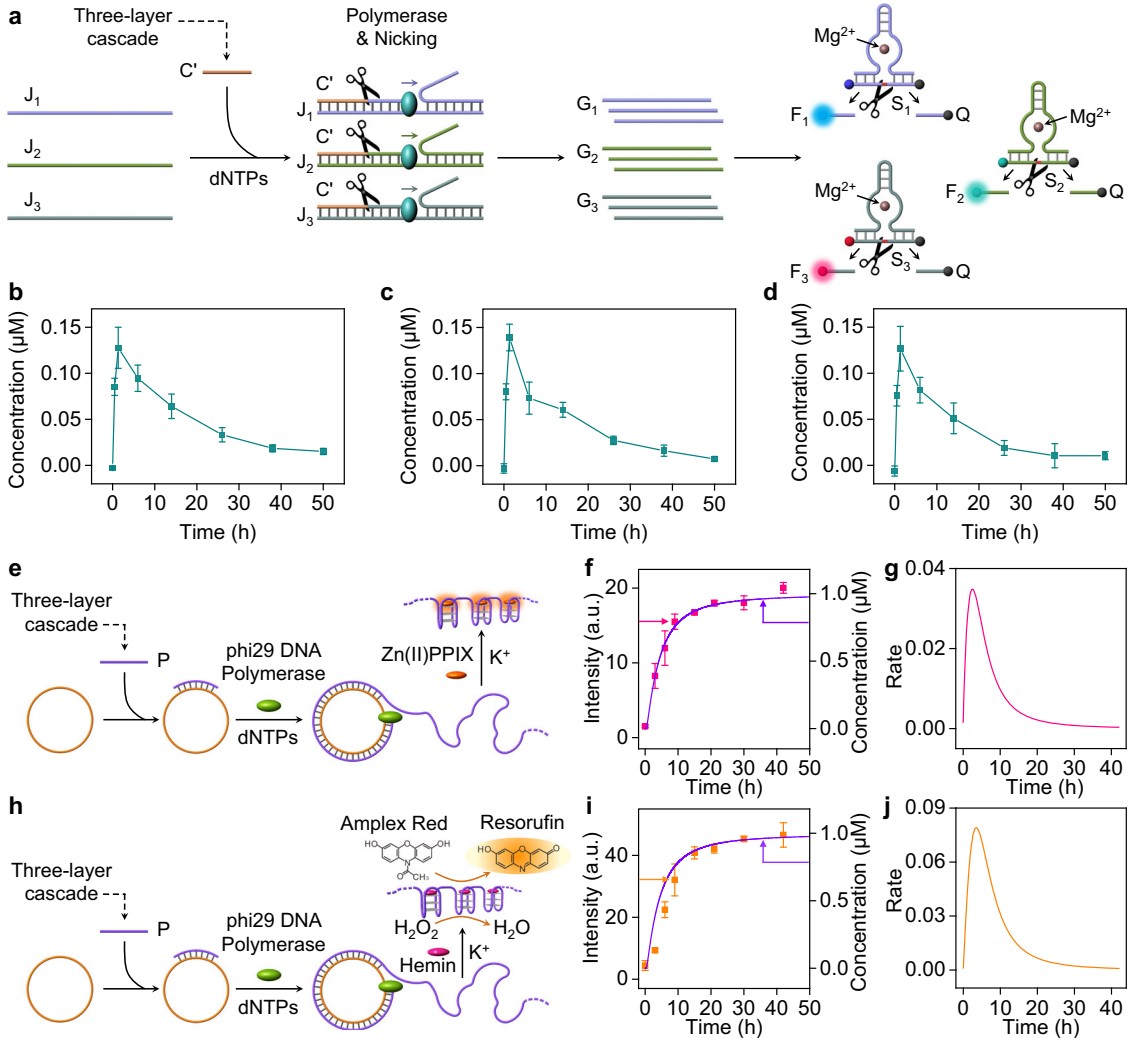

**Fig. 4 | Conjugation of the transient three-layer DNAzyme cascade to subsequent biocatalytic processes. a** Coupling of the three-layer cascade to a polymerase/nicking machinery consisting of promoter-triggered three scaffolds $J_1$-$J_3$ yielding in the presence of dNTPs, the transient synthesis of three DNAzyme coded strands, $G_1$-$G_3$. The dynamic formation of the coded strands is followed by the fluorescence changes generated by the DNAzyme catalyzed cleavage of the substrates $S_1$-$S_3$. Transient concentration changes of the DNAzyme oligonucleotides: (**b**)-DNAzyme $G_1$, (**c**)-DNAzyme $G_2$, (**d**)-DNAzyme $G_3$. **e** Coupling of the three-layer cascade to the RCA process yielding Zn(II)-PPIX-functionalized fluorescent G-quadruplex wires. **f, g**, (**f**)-Time-dependent fluorescence intensities of the Zn(II)-PPIX-generated G-quadruplex wires (square dots), and time-dependent concentrations of the promoter strand P generated by the cascade (purple solid curve).

(**g**)-Transient rates corresponding to the formation of the fluorescent Zn(II)-PPIX/G-quadruplex wires (Derivative of square dots kinetic pattern shown in **f**). **h** Coupling of the three-layer cascade to the RCA process yielding the hemin/G-quadruplex DNAzyme wires catalyzing the $H_2O_2$-induced oxidation of Amplex Red to the fluorescent Resorufin. **i, j**, (**i**)-Time-dependent fluorescence intensities generated by the Resorufin associated with the hemin/G-quadruplex-catalyzed oxidation of Amplex Red by the hemin/G-quadruplex DNAzyme wires (square dots), and time-dependent concentration changes of the promoter P generated by the three-layer cascade (solid purple curve). (**j**)-Transient rates corresponding to the formation of the hemin/G-quadruplex biocatalytic module oxidizing Amplex Red (Derivative of the kinetic pattern of the square dots shown in **i**). Error bars for different systems derived from $N = 3$ experiments. Source data are provided as a Source Data file.

conjugated to the three-layer cascade. The scaffolds $J_1$, $J_2$ and $J_3$ are engineered to act as dynamic machineries to generate three different coded strands, $G_1$, $G_2$ and $G_3$. The formation of the hybrids between the promoter C′ and $J_1$, $J_2$ and $J_3$ initiates, in the presence of polymerase and the nicking enzyme (Nt.BbvCI), the replication of the scaffold templates and the generation of the three coded strands acting as "model genes" $G_1$, $G_2$ and $G_3$. The three coded strands $G_1$, $G_2$ and $G_3$ are designed to include three different $Mg^{2+}$-ion-dependent DNAzyme sequences that cleave the respective substrates $S_1$, $S_2$ and $S_3$ comprising of $F_1$/Q, $F_2$/Q and $F_3$/Q labeled DNA sequences, respectively. The evolving time-dependent formation of the fragmented $F_1$-, $F_2$- and $F_3$-substrates provide quantitative output signals that follow the formation and catalytic function of the resulting strands. The formation of the coded strands $G_1$, $G_2$ and $G_3$ is controlled by the concentration of promoter C′. As the catalytic rate corresponding to the formation of C′

follows the dissipative transient pattern of the three layers cascade, the formation of the three strands will follow the same pattern. Accordingly, the mixture of the machineries $J_1$, $J_2$ and $J_3$ was subjected to samples of the three-layer cascade at time-intervals of operation of the three-layer cascade. Thus, the transient formation of the coded strands activated by the polymerase/nicking machineries will follow the transient kinetic features of constituent C′, predicted by the kinetic model of the three-layered cascade (see Supplementary Fig. 36). Figure 4b–d show the transient formation of the three catalytic strands $G_1$, $G_2$ and $G_3$ (for further experimental details see Methods section, and Supplementary Figs. 37–42). That is, the constituent C′ formed within the three-layer cascade guides a transient formation of the three-model genes, in analogy to native gene replication pathways. To demonstrate the functions of the transient cascade on the coupled temporal behavior of replicated three coded strands $G_1$, $G_2$ and $G_3$ and their

resulting DNAzymes activities, we subjected the scaffolds $J_1$, $J_2$ and $J_3$ to a constant concentration of C', 0.05 μM, and operated the dNTPs/polymerase/nickase replication of the genes while following their temporal formation and DNAzyme activities. The results are presented in Supplementary Figs. 43–46. One may realize that under these conditions, the formation of the strands and their catalytic features stay constant, indicating that the cascaded network, indeed, controls the transient concentrations of the genes and their catalytic functions. Furthermore, the transient operation of the three-layer cascade does not only dictate the dissipative transient emergence of coded oligonucleotides (DNAzymes), but by altering the conditions of operating the three-layer cascade the coupled transient replication of the artificial "gene model" strands (DNAzymes) can be modulated. For example, by altering the concentrations of the trigger $T_1$ operating the three-layer cascade from 3.6 μM to 1.8 μM, the peak concentration of the resulting promoter C' changed from 0.075 μM to 0.06 μM (cf. Fig. 3d, h). The consequences of lowering the concentration of C' on the dissipative replicated DNAzyme strands are displayed in Supplementary Figs. 47–50 and the yield of the resulting transient catalytic coded strands is lower. In a further control experiment, we subjected the polymerization/nicking machinery to samples withdrawn from the transient two-layer cascade that does not yield the C' product. The system did not yield any traceable DNAzyme-coded products (Supplementary Figs. 51–54), implying lack of formation of parasitic products. It should be noted that we coupled the three-layer cascade to three different polymerization/nicking machineries $J_1$, $J_2$, $J_3$ as a proof-of concept demonstrating that a dynamic three-layer cascaded network that yields a common promoter C' can activate the multiple synthesis of diverse functionally coded strands. In principle, the polymerization/nicking machineries could be activated by a single layer dissipative construct. We feel, however, the enhanced complexity introduced by the three-layer triggered operation of three different functional coded strands, as outputs, provides means to control the rates of the transient synthesis of functional oligonucleotides through regulating the coupler units. Furthermore, we note that our transient three-layer dissipative cascade that recovers to the parent state through the dissipative depletion of the transient intermediates leads to the emergent transient replication of three catalytic strands, a phenomenon that cannot be achieved under equilibrium conditions (e.g. Supplementary Figs. 43–46).

Furthermore, the promoter P generated by the three-layered cascade was applied to trigger the rolling circle amplification (RCA) process, where the circular template was engineered to generate three G-quadruplex units in each revolution of the RCA process, Fig. 4e. That is, the RCA process yields wires consisting of G-quadruplexes that in the presence of Zn(II)-protoporphyrin IX (Zn(II)-PPIX), yield fluorescent wires[59], or in the presence of hemin yield the hemin/G-quadruplex DNAzyme wires, catalyzing the $H_2O_2$ oxidation of Amplex Red to the fluorescent Resorufin (Fig. 4h). Figure 4f shows the time-dependent fluorescence intensities (square dots) of the RCA-generated Zn(II)-PPIX-functionalized G-quadruplex wires (Supplementary Fig. 55). For comparison, the purple curve overlaid on the square dots corresponds to the time-dependent concentrations of the promoter P, generated by the three-layer cascade. The concentrations of P were evaluated by the kinetic model (Supplementary Fig. 33). The dynamic fluorescence intensities of the Zn(II)-PPIX wires follow the dynamic concentrations of the promoter P, generated by the transient three-layer cascade. The time-dependent fluorescence intensities of the Zn(II)-PPIX-functionalized wires were, then, translated into the rates of the formation of the wires by the system, Fig. 4g. Evidently, the dynamics of formation of the wires shows a transient, kinetic profile. Similarly, the fluorescence intensities of the hemin/G-quadruplex catalyzed production of Resorufin by the RCA-generated DNAzyme wires are shown in Fig. 4i, square dots (Supplementary Fig. 56), including the overlaid dynamic concentrations of the promoter P generated by the three-layer cascade (solid curve). As before, the fluorescence

intensities generated by Resorufin, at time intervals of the RCA processes, are translated into time-dependent catalytic rates of the resulting hemin/G-quadruplex wires that follow a transient pattern, Fig. 4j. That is, the transient operation of the three-layer cascade is reflected in the dynamics of the coupled processes driven by the cascade.

Throughout the paper, the transient dissipative cascades were operated in pure buffer solutions. To demonstrate, however, the relevance of such systems for future applications in biological environments, it is important to demonstrate the feasibility to operate the transient reaction moduli in native environment. Accordingly, we examined the $Mg^{2+}$-ion-dependent DNAzyme, transient single-layer, double-layer and three-layer systems in epithelial breast MCF-10A cell lysate and MDA-MB-231 breast cancer cell lysate. We find that the cascaded networks retains their activations in the cell lysates for three days. The results are presented in Supplementary Figs. 57–60, and these demonstrate the feasibility to operate the systems in biological environments.

## Discussion

Extensive advances were recently reported with the development of dynamically reconfigurable equilibrated constitutional dynamic networks[60,61]. In addition, recent advances demonstrated the assembly of dynamically transient, out-of-equilibrium, nucleic acid reaction moduli driven by an enzyme (nickase) as control element of the temporal reactions. The present study introduces DNAzymes as constituents acting concomitantly as the transient reaction constituents and as the control elements of temporal processes. This dual activity of the DNAzyme allows the enhancement of the complexity of the transient process by guiding cascaded temporal processes that are coupled to synthesis of functional coded oligonucleotide strands using replication routes. The study demonstrated the assembly of a cascaded transient multi-layer network driven by catalytic nucleic acids. The cascaded dissipative network was conjugated to DNA machineries for the guided synthesis of functional coded oligonucleotides, acting as "gene models", and for the activation of the RCA processes synthesizing catalytic nucleic acid chains. While the results demonstrate the assembly of functional transient network systems of enhanced complexities, the results pave concepts for future advances. At present, we made use of the $Mg^{2+}$-ion-dependent DNAzyme as catalyst to drive the cascaded network. The availability of many other cofactor dependent DNAzymes suggests that additional networks could be constructed. In addition, the intercommunication between the cascaded layers generated "waste" products ($H_{1b}$/S; $H_{2b}$/M and $H_{3b}$/W). These waste products could act as precursors triggering branched networks and cascaded branched networks, thus enhancing the complexities and functionalities of the transient, dissipative, networks. Furthermore, at present, the layered network consists of all-DNA constituents, yet recent studies demonstrated the benefits of conjugated hybrids composed of native enzymes linked to dynamically reconfigured DNA networks[48]. By coupling enzymes to the nucleic acids participating in the layered transient cascades, complex transient biocatalytic transformations, eventually for therapeutic applications, may be envisaged. Also, the output strand, C', used to trigger the transient formation of the "coded strands" could act as trigger to operate transient logic gates or computing circuits[62–64], thus providing a new dynamic element to the area of DNA computing. While preliminary studies demonstrated the feasibility to perform the dissipative networks in native fluids, the future integration of the systems in cells or protocell assemblies[65], such as liposomes, polymersomes or dendrosomes could be important paths to follow.

## Methods
### Materials
Oligonucleotides were obtained from Integrated DNA Technologies (IDT). Optimized oligonucleotide structures were derived using

NUPACK (version 4) software[66]. The fluorophore- and quencher-labeled oligonucleotides were purified by high-performance liquid chromatography. All the sequences of the oligonucleotides are listed in Supplementary Table 7. Klenow Fragment (3′→5′ exo-) (M0212L), Nt.BbvCI (R0632L), deoxyribonucleoside 5′-triphosphate mixture (dNTPs, 10 mM), T4 DNA ligase (M0202S), Exonuclease I (Exo. I, M0293S), Exonuclease III (Exo. III, M0206S), BSA (10 mg ml$^{-1}$) and phi29 DNA polymerase (M0269S) were obtained from New England Biolabs (NEB). All other chemicals were obtained from Sigma-Aldrich. Ultrapure water produced by NANOpure Diamond (Barnstead) was used in all the experiments.

### Characterization of the single DNAzyme dissipative module
Taking module I as an example. A volume of 150 μL consisted of a mixture that included AA′ (0.2 μM), H$_1$ (1 μM) in Tris-Mg buffer (10 mM Tris, 20 mM MgCl$_2$, pH 7.4). The hairpin strands and the duplex strands were prepared in Tris-Mg buffer (10 mM Tris, 20 mM MgCl$_2$, pH 7.4) and annealed from 90 °C to 10 °C at a rate of −1 °C min$^{-1}$ before mixing with other strands. Variable concentrations of the trigger strand, T$_1$, (0.2, 0.3 and 0.4 μM) were added into the solution comprising the module and the fluorescence changes were recorded with a Cary Eclipse Fluorescence Spectrophotometer (Varian, Inc.) ($\lambda_{ex}$ = 635 nm, $\lambda_{em}$ = 665 nm). For the cyclic measurement, the trigger strand, T$_1$ (0.4 μM), was repeatedly added to the system.

### Characterization of bilayer DNAzyme transient cascaded processes
Taking the cascaded transient system consisting of module I and II as an example. The coupler duplex, S/T$_2$ (1.2 μM or 1.8 μM), was mixed with the constituents composed of AA′ (0.6 μM), H$_1$ (3 μM), BB′ (0.2 μM) and H$_2$ (1 μM) dissolved in 150 μL Tris-Mg buffer (10 mM Tris, 20 mM MgCl$_2$, pH 7.4). Variable concentrations of trigger T$_1$, (1.2 and 1.8 μM) were added into the solutions and the fluorescence changes were recorded with the Cary Eclipse Fluorescence Spectrophotometer (Varian, Inc.) (Cy5, $\lambda_{ex}$ = 635 nm, $\lambda_{em}$ = 665 nm; FAM, $\lambda_{ex}$ = 495 nm, $\lambda_{em}$ = 518 nm). For the cyclic measurement, the trigger strand, T$_1$ (1.2 μM), was re-added to the mixture.

### Characterization of three-layer DNAzyme transient cascades
The duplexes, S/T$_2$ (1.8 μM or 3.6 μM), M/T$_3$ (0.6 μM or 1.2 μM) and W/P (2 μM), were mixed with a solution that included AA′ (1.8 μM), H$_1$ (9 μM), BB′ (0.6 μM), H$_2$ (3 μM), CC′ (0.2 μM) and H$_3$ (1 μM) in 150 μL Tris-Mg buffer (10 mM Tris, 20 mM MgCl$_2$, pH 7.4). Variable concentrations of trigger strand, T$_1$, (1.8 and 3.6 μM) were added into the solutions and the fluorescence changes were recorded (Varian, Inc.) (Cy5, $\lambda_{ex}$ = 635 nm, $\lambda_{em}$ = 665 nm; FAM, $\lambda_{ex}$ = 495 nm, $\lambda_{em}$ = 518 nm; Cy3, $\lambda_{ex}$ = 545 nm, $\lambda_{em}$ = 565 nm).

### Coded strands replication guided by transient cascade
Samples consisting of 30 μL solution of the operating three-layer cascaded system (trigger T$_1$, 3.6 μM were added) at variable intervals of the operating cascade were added into a mixture 100 μL that included the three DNA templates (0.01 μM each), 2 μL Klenow Fragment (3′→5′ exo-), 2 μL Nt.BbvCI and dNTPs (0.3 mM) in 1× NEBuffer™ 2 (50 mM NaCl, 10 mM Tris-HCl, 10 mM MgCl$_2$, 1 mM DTT, pH 7.9) to react at 28 °C for 6 h. Subsequently, the solution was treated with the substrates S$_1$, S$_2$ and S$_3$ (2 μM each) and the time-dependent fluorescence of the respective fluorophore labeled fragmented substrates were followed (Cy5, $\lambda_{ex}$ = 635 nm, $\lambda_{em}$ = 665 nm; FAM, $\lambda_{ex}$ = 495 nm, $\lambda_{em}$ = 518 nm; ROX, $\lambda_{ex}$ = 588 nm, $\lambda_{em}$ = 608 nm). The concentrations of the coded DNAzyme strands were evaluated at different time-intervals of operation of the three-layer cascade. Each data point was evaluated by probing the catalytic activity of the respective DNAzyme and using the appropriate calibration curve (for further details see Supplementary Discussion, page 47).

### Dynamic RCA processes guided by transient three-layer cascade
The circular template was prepared by capping the linear strand T$_R$ by the strand C$_L$ (2 μM each, 100 μL). The mixture was annealed from 60 to 10 °C at a rate of −0.4 °C min$^{-1}$ in 1× T4 DNA ligase reaction buffer (50 mM Tris-HCl, 10 mM MgCl$_2$, 1 mM ATP, 10 mM DTT, pH 7.5) followed by the addition of 5 μL T4 DNA ligase, and then the mixture was allowed to react for 12 h at 25 °C. Subsequently, 1 μL Exo. I and 2 μL Exo. III were added to the mixture to digest the linear DNA strands for 2 h at 25 °C followed by heating the solution to 80 °C for 20 min to denature the enzymes. The circular template was purified by PAGE electrophoresis.

Samples of 5 μL solution of the three-layer operating cascade system (trigger T$_1$, 3.6 μM), at time intervals of operating cascade, were added to the RCA solution machinery that included the circular template strand (5 μL), dNTPs (0.5 mM) and phi29 DNA polymerase (2 μL) in 1× phi29 DNA Polymerase Reaction Buffer (200 μL, 50 mM Tris-HCl, 10 mM MgCl$_2$, 10 mM (NH$_4$)$_2$SO$_4$, 4 mM DTT, pH 7.5) including BSA (0.2 mg mL$^{-1}$) and K$^+$ ions (20 mM). The solution was kept at 28 °C for 30 min followed by heating to 65 °C for 10 min. For the dynamic formation of the fluorescent G-quadruplex wires, a solution of Zn(II)-PPIX (10 μM) was added to the RCA machinery solutions and the fluorescence of the product was followed at $\lambda_{ex}$ = 420 nm, $\lambda_{em}$ = 500 – 750 nm. For the dynamic formation of the DNAzyme G-qudruplex/hemin catalytic wires, the hemin (10 μM), Amplex Red (400 μM) and H$_2$O$_2$ (5 mM) were added to the RCA machinery samples. The mixtures were incubated at 28 °C for 20 min and the fluorescence of the Resorufin product was followed at $\lambda_{ex}$ = 571 nm, $\lambda_{em}$ = 580–750 nm.

### Reproducibility and error-evaluation of the transient curves
All transient curves presented in the study were reproduced by $N$ = 3 experiments. The temporal curves demonstrated a ≤ 3% deviation in the different experiments.

### Computational kinetic simulations of the experimental results
Kinetic models for each of the single-layer, bi-layer and three-layer systems were formulated. The initial concentrations of the constituents used in the simulations of the different systems are provided for each of the respective kinetic models. The simulation processes are constrained by the law of mass conservation of the intermediates as a prerequisite, and performed using the Matlab R2020a software. The set of the rate constants associated with the optimized fitted curves were evaluated. To support the derived set of rate constants as a meaningful representative solution for the system (rather than a coincidental local solution), we used this set of rate constants to predict the behavior of the system at different auxiliary conditions and experimentally validated the predicted results. In addition, each set of simulated rate constants was supplemented by two experimental rate constants (total six rate constants for the three-layer systems). The experimental rate constants were in good agreement with the simulated values. The tables corresponding to the rate constants associated with each of the systems are included in the Supplementary Tables 1–6 and the respective red values in squares correspond to the experimentally validated values. (For the experimental results of these supportive values of rate constants, see Supplementary Figs. 4, 10, 15.) Note, however, that although the single-, two- and three-layer systems include common differential rate equations, the derived simulated rate constants for these reactions in the respective networks differ in their values. This is attributed to the fact that the initial concentrations of the constituents and, thus, the constraints of mass conservations in the respective systems are different.

The time-dependent concentration changes of the separated fluorophore-labeled strand in module I, Fig. 1, dA/dt, are monitored experimentally by the total transient fluorescence intensities associated with DNAzyme T$_1$/A (including the free DNAzyme T$_1$/A and the T$_1$/A transient binding with H$_1$ and the cleaved H$_1$). For the simulation of the transient curves, we derive the time-dependent concentration

changes of A′, dA′/dt, that equal to the fluorescence time-dependent changes corresponding to dA/dt. Similarly, the time-dependent concentration changes of the separated fluorophore-labeled strands generated by module II and module III (Figs. 2, 3, and Supplementary Figs. 7, 12 and 24), dB/dt and dC/dt, are monitored experimentally by the total transient fluorescence intensities associated with DNAzyme $T_2$/B (including the free DNAzyme $T_2$/B and the $T_2$/B transient binding with $H_2$ and the cleaved $H_2$) and with DNAzyme $T_3$/C (including the free DNAzyme $T_3$/C and the $T_3$/C transient binding with $H_3$ and the cleaved $H_3$), respectively. For the simulations of the transient respective curves, we derive the time-dependent concentration changes of B′, dB′/dt, and of C′, dC′/dt that equal to the values of dB/dt and dC/dt, respectively.

## Reporting summary

Further information on research design is available in the Nature Research Reporting Summary linked to this article.

## Data availability

All Data supporting the findings of this paper are available in the main text and/or in the Supplementary Information. Source data is available for Figs. 1b, c, 2b–g, 3b–d, 3f–h, 4b–d, 4f-g and 4i-j and Supplementary Figs. 3b, 5, 6, 7b–e, 11, 12b–e, 16, 19–23, 24b–d, 25, 28–31, 34, 35, 38b, 40b, 42b, 43, 47, 51, 57–59 and 60g–i in the associated Source Data file. Source data are provided with this paper.

## Code availability

Custom code used in this study is available from the corresponding author upon request.

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

## Acknowledgements

The research is supported by the Israel Science Foundation (Grant No. 2049/20).

## Author contributions

J.W and Z.L designed and performed the experiments, analyzed the results and participated in formulating the paper. I.W mentored the project and participated in writing the paper.

## Competing interests

The authors declare no competing interests.
