## [Peer Review File · Nature Communications]

REVIEWER COMMENTS

Reviewer #1 (Remarks to the Author):

The work presented by Wang and coauthors describes three DNA-based cyclic reactions that are linked in a sequential temporal manner. The final product of the cascade is used to trigger the amplification of DNA sequences by means of polymerase and nicking enzymes. The accumulation of these "model genes" is monitored by fluorescence spectroscopy and the signal is associated to the formation and biocatalytic activity of DNAzymes or G-quadruplex wires. Each cyclic reaction of the cascade (called module) consists of a duplex species (AA', BB', CC') that is converted into a DNAzyme (AT1, BT2 or CT3) upon addition of a trigger DNA (T1, T2 or T3). The solution contains a large excess of hairpin and magnesium ions, so that the DNAzyme species, once it is formed, cleaves the hairpin substrate, producing waste and reconstituting the initial duplex state. The DNAzyme has therefore a transient life. The waste of each cycle is then used to liberate a different trigger strand from another duplex, thus initiating the cyclic reaction of a different module. The authors analyze the cascade of three consecutive reactions by ensemble FRET spectroscopy and perform kinetic simulations of the experimental curves to describe the single and connected modules.

The work is clearly done and particularly interesting as it touches the modern topic of dissipative structures, an actual field in systems chemistry.

1. The authors describe their transient (metastable) system as dissipative. As also reported by other authors, particularly from the field of supramolecular chemistry, systems of this type dissipate the energy stored in their chemical bonds by reverting to an initial precursor, which in turn occupies a global minimum in the potential energy surface. In other disciplines, such as supramolecular polymerization and mostly systems biology, dissipative structures essentially exist only on the slope of the energy diagram and can maintain their dynamic instability only by continuous supply of energy or matter. For the sake of clarity, I would suggest the authors to briefly define the context in which their system is defined as dissipative.

My major concerns are however related to the use of the energy stored in the metastable species and the added value of this study when compared to previous works from the same group (points 2 and 3 below). Some minor points are also given below and are intended to improve the manuscript and Supplementary Materials in clarity and details.

Major points:

2. Typically, the reason for creating such dissipative structures is to employ the energy stored in the metastable state for some purpose. In nature (and in the original sense given by Prigogine), dissipative structures lead to emergent properties, such as oscillations and spatio-temporal patterns that cannot be formed in equilibrium conditions. To better mimic this property of dissipative structures, the metastable species is often coupled to a self-assembly process and observed in a sustained condition of energy or matter supply. The authors use instead the last released strand of the third cycle (strand C') or the final released waste product (strand P) to trigger a fourth DNA polymerase reaction. I am not sure to see the advantage of using a transient process to perform the target reactions. The transient dissipative nature of the system is in my opinion not really exploited, rather observed, and instead the waste is used to trigger another cycle.

Maybe to prove or disprove this concept, the authors should show what happens if the polymerase reaction depicted in figure 4a is fueled by a defined concentration of C', however, in absence of the entire cascade. This may better evidence the impact of the cascade and the advantage of having it. It would be also helpful to state if the substrate molecules S1, S2 and S3 are already present in solution and at which concentration. To my understanding, in presence of magnesium ions and excess substrate molecules, G1, G2 and G3 sequences form transient DNAzymes, according to a mechanism that is very similar to the one displayed by the DNAzymes of the cyclic reactions. If this is true, then it would not be surprising to observe a transient behavior of activity, even in absence of the cascade.

3. The system proposed by the authors can be surely claimed to mimic the complexity of linked reaction cascades, and it does undoubtedly well. However, the authors showed already a very similar work in JACS 2021 (ref. 57). The improvement presented in this study relies on the coupling of a third cycle to a DNA-polymerase reaction. Can the authors better strengthen the significance of this addition? can for example the kinetics of the cascades be changed in a way to control the dynamics of the final event (see also my point above)? It is otherwise difficult to appreciate the new learning message of this work.

Minor points:

4. kinetic models. All rate constants given in this work are the solution of systems of several ODEs. More details about the simulation procedure should be given, such as the initial conditions of the system and the law of mass conservation.

5. Most importantly, it should be stated whether the same simulation conditions have been maintained within the same module. For example, in module I, are the 13 rate coefficients of a' , really the same as in b' and c' ? what does it mean exactly "best-fitted computational curve"? To me, it sounds like the parameters have been optimized (manually?) till the simulated curves mostly approached the experimental curves. Has this been done for all curves simultaneously? like in a global fit procedure?

6. Did the authors try to corroborate the given values of the rate coefficients by performing a global fit of all experimental curves of the same module, using the simulated values as starting parameters? In general, I like very much the mathematical treatment, it is nice and elegant, but one should also admit the realistic limitations of this mathematical treatment that predicts up to 48 rate coefficients on the basis of a limited number of curves.

7. The authors should clearly state that these are batch-fueled cycles, meaning that the operations of the modules are triggered by consecutive additions of fuels, or as long as fuels are available. How many cyclic operations are done for each module?

8. please also state the half-life of the DNAzyme decay in each cycle.

9. The authors use the term "cascadability". Maybe "ordered connectivity" can be an alternative?

10. page 2 to 4 of the introduction: I suggest to improve the English form here. Some sentences are unnecessarily complicated or too long and make an excessive use of the "ing" form.

11. p5: "The reaction module I consists of the toehold-functionalized duplex A/A' where strand A is modified at its 5' end with the fluorophore Cy5 and strands A' at its 3' end with the quencher BHQ2, the hairpin H1 and Mg^{2+} ions."

I believe in the following way, the message will become clearer: "The reaction module I consists of the toehold-functionalized duplex A/A' , the hairpin H1 and Mg^{2+} ions. Strand A is modified at its 5' end with the fluorophore Cy5 and strands A' at its 3' end with the quencher BHQ2.

12. p5: "Through the displacement of the quencher-modified strand A' , leading to the switched "ON" the fluorescence of the duplex $T1/A$." This sentence is not fully clear to me.

13. Fig. 1: the initial concentration of A/A' and H1 species should be given in the legend to appreciate the kinetics of the reaction.

14. Also, in Fig. 1c, it should be made clear that the second cycle starts upon external addition of T1.

15. p6-7. "Very good fit between the predicted and experimental results is observed, demonstrating the value of the predictive model to follow the transient behavior of the system." This claim should be in my opinion more cautiously addressed. Although the authors have already amply provided evidence of being able to realize chemical reaction networks in different conditions, I believe it would be extremely useful to add here a non-denaturing gel

characterization of the species present in solution at different times. This should help to visualize transient species which are not fluorescently labelled and might be also of help to validate the rate coefficients simulated. Indeed, the Matlab code can be easily modified to generate the time profile of any desired species of the ODE system. Such theoretical predictions can be easily compared with the appearance/disappearance and relative intensity of the gel bands associated to the distinct intermediate species. This would strongly support the theoretical model.

16. p. 11: "...increasing the concentration of the trigger T1 intensifies the peak content of T2/B and enhances the time for generating the peak content to 108 minutes.." probably a typo, as it shortens the time, i.e. enhances the rate ...

17. Suppl. Fig. 3, 8 and 13: the simulations for the module I and particularly III are – in my opinion - not well corresponding to the experimental data points. Module II instead is fitting very well. Can the authors comment on this?

18. Also, I would suggest to improve the nomenclature of the simulated/experimental curves. In the suppl. figures 3, 8 and 13, a/a' and b/b' refer to a different ratio of the duplex strands in the module. In other suppl. figures and in the main manuscript, the a/a' , b/b' and c/c' curves refer to different conc of trigger (T1, T2 or T3). Furthermore, the modules are made with duplexes A/A', B/B' and C/C' which makes all more complicated to follow.

19. fig. 4: d and e: in panels I, the experimental dots are fitted with a hyperbolic function. In panels II, the rate of the reaction is plotted vs time. The graphs of panels II are not the first derivative of the hyperbolic function shown in panels I, rather the first derivative of the "sigmoidal" shape followed by the experimental points. This clearly better evidences the transient feature of the system's response. Nevertheless, it is not clear to me why the fits in panels I are done with a hyperbolic function and the 1st derivatives in panels II are not done on the fits. So what is the kind of function that actually better describes the experimental points? is it a hyperbolic or sigmoidal function?

20. the sequences of the "G genes" are not provided in the Supplementary materials.

Reviewer #2 (Remarks to the Author):

Present manuscript by Willner et al. aims to establish a DNAzyme initiated three layers cascaded non equilibrium reaction network using nucleic acid strand displacement reactions in combination with DNAzyme type restriction. An important class of dynamic reaction network mimicking biological complex reaction cascades are DNA based transient signal triggered systems. Development over the years have led to oscillatory behaviour, bistability, fueled out of equilibrium ligation cycles, transient formation of DNA fibers etc. Particularly, complex chemical circuits mimicking natural chemical reaction networks were achieved. The important challenges regarding future design of DNA based chemical reaction network consist of connectivity between reaction network and cascability. Herein, the authors based on their previous expertise of transient DNAzyme based non-equilibrium reaction network have demonstrated interconnectivity between three chemical reaction modules. The fueled activation of a dormant module activates a DNAzyme which initiates three layered cascades of DNAzyme reaction network. Finally, to imitate cascaded reaction network driven function in biology authors have coupled it to gene generating machineries or an RCA machinery. Overall, the manuscript demonstrates a nice concept of cascability and interconnected DNA based non-equilibrium reaction networks. The authors have previously established constitutional dynamic network using DNA based systems and here they present a step forward towards interconnected chemical reaction network. Previously cascaded DNA based reaction networks with two layers of cascability has been reported (J. Am. Chem. Soc. 2021, 143, 5071–5079, J. Am. Chem. Soc. 2020, 142, 21102–21109, J. Am. Chem. Soc. 2020, 142, 17480–17488). The present manuscript definitely elevates the complexity, particularly coupling of the output of three layers of cascaded reaction network to function is intriguing and the present

manuscript can be accepted in nature comm, pending some revisions. Some of the comments about the manuscript is as follows.

– Figure 1a authors have shown the amount of DNA zyme formed on addition of the trigger. But the curve never reaches its initial position and the deviation from the initial state increases with increase of trigger concentration. I guess these must be due to some unwanted interaction between the DNAs. Can the authors shed light on some of the side reaction taking place during the reaction network?

– The authors say that the recovery rate of the rest module slows down with increase of trigger concentration. I can not see a curve in supporting information which quantitatively plots the differences of their lifetime. I suggest the authors to include a figure in SI which shows differences of their lifetime.

– The fits reported in Supplementary Figure 3 are poor. I suggest to cut down the saturated portion and fit the data again to get the correct values.

– Can the author also include the blank where addition of hairpin H1 does not initiates layer 2?

– In figure caption figure heading are a bit confusing. For example, in Figure 1b the authors have defined three curves with ((a)/(a'), (b)/(b'), (c)/(c')) however there is a separate Figure 1c is there which is quite confusing and you have to read the figure caption many times before you get it. I suggest the authors to use different number so that it gets clearer. Also please mention concentration of the duplex and hairpin in the figure caption. Similarly supplementary figures should also include concentration of the species used during the study for better readability.

– In figure 2d why the second cycle shows an increased yield? A higher concentration of trigger was added for the second cycle? One does not know by reading the figure caption or main text. Also, what is the difference in lifetime during second cycle? Corresponding data to quantify the lifetime also should be there in SI.

– In the reaction network design, the two layers are coupled using module S/T2. I wonder what is the advantage of using a module in between? Why the released H2 cannot directly participate in the second layer?

– For the three-layer cascaded system, the second layer reaches peak value earlier (108 minutes in Figure 2 and 35 minutes in figure 3). What is the reason? Again, is it due to different concentration?

– The caption of Supplementary Figure 27 reads “The transient formation of the constituent C' predicted by the three-layered cascade.” Predicted by what? There is no accompanying discussion to understand the experiment.

– Supplementary Figure 26 shows a control where absence of duplex P/W does not activate the third layer of reaction network. However, the fluorescence data shows some decrease. What could be the reason behind that?

– The authors have utilized DNAzyme activity to measure the transient generation of genes G1, G2 and G3. I wonder why the corresponding time dependent fluorescent changes are linear (Supplementary Figure 28, 29, 30, 31 etc)? The curves do have any saturation and how do the authors calculate the rate from that? Since it is a catalytic reaction, I am wondering they should be exponential curves rather than linear?

– What was the purpose of coupling it to three gene replication machineries? I guess one would have sufficient to show the concept. We do not get any extra information by coupling it to three gene replication machinery.

– Also, it is not so clear to me why the concentration of the gene G1, G2, and G3 changes transiently? As I can understand from the scheme the formed genes G1, G2, and G3 does not undergo any kind of degradation. So, I wonder why the concentration of the DNAzyme will decrease transiently in figure 4b? Does the Figure 4b means rate of catalytic action of the DNAzymes?

– Important: if this is not the G1, G2, G3 concentration, then please measure those concentrations.

– What is the extent of parasitic activation in these Polymerase/Nickase reactions and in the RCA. Especially Pol/Nic networks are prone to parasitic activation, meaning G1,G2,G3 will be produced no matter whether the C' Cascade output is there. Provide measurements

– I would also expect the gene replication machinery to reach the peak value later than the three layers of cascade. Can the author co plot them to see that the gene replication machinery is delayed compared to the three layers of connected cascade?

– Important: Almost all data appears to be single point measurements. At least duplicates need to be provided.

→ Important modeling: Make a table listing all experimentally determined parameters and listing all fitted parameters for all reaction networks. Otherwise no one can understand the model or even reproduce parts of it. It appears that the models use many many fitting parameters. The ratio of determined values, vs fitted parameters needs to be discussed. If the ratio is completely off on the fitted values, the fits are not really meaningful as there can be so many local solutions to this problem.

Reviewer #3 (Remarks to the Author):

Wang et al presented an interesting paper showing that a transient three-layer DNAzyme cascade network could be fabricated and used to trigger biocatalytic processes. This DNAzyme cascade strategy displayed transient model gene synthesis and RCA process that yielded fluorescent G-quadruplex wires. However, to meet the high criteria of Nature Communications, the authors should thoroughly address the following concerns.

1 Similar DNAzyme-based concept has been reported many times in previous publications of the same group (JACS, 2021, 143, 42, 17622; JACS, 2021, 143, 31, 12120; JACS, 2021, 143, 13, 5071; JACS, 2021, 143, 1, 241; JACS, 2020, 142, 52, 21577; JACS, 2020, 142, 51, 21460; JACS, 2020, 142, 41, 17480). The authors should fully describe the novelty of current manuscript.

2 In this manuscript, the DNAzyme cascade still worked in the buffer solution. Can the authors test this DNAzyme-based system in physiological environments (eg, cell lysate, living cell membranes or cytoplasm) to investigate their functions? It will be convincing and of great importance, if the authors could demonstrate their DNAzyme cascade performs triggered gene replication inside the cells. Otherwise, another proof-of-concept demonstration can not meet the high criteria of Nature Commun.

3 What is the stability of the DNAzyme cascade network in the working solution and in the physiological environments? Experimental details to examine the resistance to nuclease degradation are needed.

4 All the DNAzyme networks used in the paper are Mg²⁺-ion-dependent. Is the concentration of Mg²⁺ ions in the physiological environment sufficient to drive the DNAzyme cascade to perform the corresponding functions?

Reviewer #1

We are pleased with the general comment of the reviewer that “The work is clearly done and particularly interesting as it touches the modern topic of dissipative structures, an actual field in systems chemistry.”

The following changes were introduced into the paper to address the comments of the reviewer.

1. “.....*For the sake of clarity, I would suggest the authors to briefly define the context in which their system is defined as dissipative.*”

Reply: This issue is now explicitly addressed in the introduction section, page 5 by explaining that we consider each of the dissipative DNAzyme layers as a supramolecular reaction module that is activated by the supply of energy in the form of a fuel strand that generates waste products and a non-equilibrated reaction intermediate that undergoes transient depletion to the initial self-assembled reaction module.

2. “.....*I am not sure to see the advantage of using a transient process to perform the target reactions. The transient dissipative nature of the system is in my opinion not really exploited, rather observed, and instead the waste is used to trigger another cycle.*”
“*Maybe to prove or disprove this concept, the authors should show what happens if the polymerase reaction depicted in figure 4a is fueled by a defined concentration of C', however, in absence of the entire cascade. This may better evidence the impact of the cascade and the advantage of having it.....*”

Reply: We feel that reviewer missed the functions of the transient DNAzyme layer in controlling the modulated transient formation of the “genes” G₁, G₂, G₃. In fact, his/her suggestion to probe the transient features of C' on the emerging transient features of the genes G₁, G₂, G₃ by applying a constant concentration of C' on the emergence of the genes is excellent. This experiment was performed and is described on page 20 and the results are shown in the Supplementary Figures 41-44. Under these conditions, the formation of the genes and their catalytic functions stay constant, and thus, the transient cascaded network, indeed, dictates the emerging dissipative properties of the DNAzyme genes. This issue is further emphasized in an explicit explanation on page 21.

“*It would be also helpful to state if the substrate molecules S1, S2 and S3 are already present in solution and at which concentration. To my understanding, in presence of magnesium ions and excess substrate molecules, G1, G2 and G3 sequences form transient DNAzymes, according to a mechanism that is very similar to the one displayed by the DNAzymes of the cyclic reactions. If this is true, then it would not be surprising to observe a transient behavior of activity, even in absence of the cascade.*”

Reply: We feel that the reviewer missed the way to derive the temporal concentrations of the genes generated by the transient three-layer cascade, displayed in Figure 4b. In fact, the procedure was detailed in the original manuscript page 28 and the information requested by the reviewer was presented in that section.

For the sake of clarity, we repeat step-by-step the procedure to evaluate these concentrations:

- (i) We take samples out of the three-layer cascade at different time-intervals of operation.
- (ii) We apply the replication/nicking machinery synthesizing the genes G_1 , G_2 , G_3 on each of the samples for a time-interval of six hours.
- (iii) After completion of the synthesis of the genes, the substrates S_1 , S_2 , S_3 are added at an excess ($2 \mu\text{M}$), and the concentrations of the synthesized genes are quantitatively evaluated by following the rates of cleavage of the substrates for a short time interval (100 minutes). The derived concentrations represent the temporal behavior of the formation of the respective genes.

3. *“.....The improvement presented in this study relies on the coupling of a third cycle to a DNA-polymerase reaction. Can the authors better strengthen the significance of this addition? can for example the kinetics of the cascades be changed in a way to control the dynamics of the final event (see also my point above)? It is otherwise difficult to appreciate the new learning message of this work.”*

Reply: Indeed, as the reviewer pointed out the significance of the presented study relies on the coupling of the third cycle of the three-layer cascade to the polymerase-guided synthesis of the genes. The reviewer is certainly, correct that the dynamics of the final polymerization event should be modulated by the conditions at which the cascade is operated. This is now discussed on page 20 and an experiment demonstrating that the conditions at which the cascade operates, indeed control the final gene generation process. The experimental results are presented in Supplementary Figures 45-48.

Minor points:

4. *“More details about the simulation procedure should be given, such as the initial conditions of the system and the law of mass conservation.”*

Reply: As requested, further details on the simulation procedures are provided in the “Methods” section, page 30, and the initial conditions of the system were added to the text. The simulation procedure followed to the law of mass conservation, and this is explicitly stated in the section describing the simulation process.

5. *“Most importantly, it should be stated whether the same simulation conditions have been maintained within the same module..... Has this been done for all curves simultaneously? like in a global fit procedure?”*

Reply: For each of the reaction module associated with the one-, two- or three-layer systems, the initial concentrations of the constituents are different, and these concentrations provide the law of mass conservation constrains for the simulations of the different systems. This is explicitly stated in the simulations procedure.

6. *“Did the authors try to corroborate the given values of the rate coefficients by performing a global fit of all experimental curves of the same module, using the simulated values as starting parameters? In general, I like very much the mathematical treatment, it is nice and elegant, but one should also admit the realistic limitations of this mathematical treatment that predicts up to 48 rate coefficients on the basis of a limited number of curves.”*

Reply: We realize, as stated by the reviewer, the limitations of the simulation procedure, and we value his/her appreciation of the simulation efforts. This is indeed, a challenge! In the method sections, page 30-31, we add a detailed explanation on the simulation procedure and explain the tools undertaken by us to support the simulations (experimental validation of predicted results and experimental evaluation of a part of the rate constants of the reactions participating in the networks).

7. *“The authors should clearly state that these are batch-fueled cycles, meaning that the operations of the modules are triggered by consecutive additions of fuels, or as long as fuels are available. How many cyclic operations are done for each module?”*

Reply: Indeed, the cycles are trigged by consecutive addition of fuels. This is explicitly stated in the caption and marked with arrow. We added a comment on the number of cyclic operations, page15, and addressed the effect of cycling on the recovery of the transient systems.

8. *“please also state the half-life of the DNAzyme decay in each cycle.”*

Reply: The half-life of the DNAzyme decay transients were added in the Supplementary Figures 6-7, 12, 22, 24, 33.

9. *“The authors use the term “cascadability”. Maybe “ordered connectivity” can be an alternative?”*

Reply: We followed the suggestion of the reviewer and altered the term “cascadability” into “ordered connectivity”.

10. “page 2 to 4 of the introduction: I suggest to improve the English form here. Some sentences are unnecessarily complicated or too long and make an excessive use of the “ing” form.”

Reply: The introduction section was re-edited as requested.

11. “..... “The reaction module I consists of the toehold-functionalized duplex A/A', the hairpin H1 and Mg²⁺ ions. Strand A is modified at its 5' end with the fluorophore Cy5 and strands A' at its 3' end with the quencher BHQ2.”

Reply: The respective sentence on page 6, was changed as suggested. Indeed, it is now clearer.

12. “p5: “Through the displacement of the quencher-modified strand A', leading to the switched “ON” the fluorescence of the duplex T1/A.” This sentence is not fully clear to me.”

Reply: The respective sentence on page 6 was re-edited.

13. “Fig. 1: the initial concentration of A/A' and H1 species should be given in the legend to appreciate the kinetics of the reaction.”

Reply: As requested, the initial concentrations of AA' and H₁ were added to the caption of Figure 1.

14. “Also, in Fig. 1c, it should be made clear that the second cycle starts upon external addition of T1.”

Reply: As suggested, we emphasized on page 7 that the second cycle is initiated by the external addition of T₁.

15. “.... I believe it would be extremely useful to add here a non-denaturing gel characterization of the species present in solution at different times. This should help to visualize transient species which are not fluorescently labelled and might be also of help to validate the rate coefficients simulated. Indeed, the Matlab code can be easily

modified to generate the time profile of any desired species of the ODE system. Such theoretical predictions can be easily compared with the appearance/disappearance and relative intensity of the gel bands associated to the distinct intermediate species. This would strongly support the theoretical model.”

Reply: Although the experiment suggested by the reviewer to support the transient behavior of the systems was a challenge, we appreciate this constructive suggestion of the reviewer. This experiment was performed and is addressed in the text, page 7, and presented in the supporting information, page 4. The transient electrophoretically separated bands fit well with the predicted transient concentrations, using the kinetic model.

16. *“p. 11: “...increasing the concentration of the trigger T1 intensifies the peak content of T2/B and enhances the time for generating the peak content to 108 minutes..” probably a typo, as it shortens the time, i.e. enhances the rate ...”*

Reply: Indeed, the reviewer is correct. It was a typo error. “Increasing the concentration of the trigger T₁ intensifies the peak content of T₂/B and **shortens** the time.....” This was corrected on page 12.

17. *“Suppl. Fig. 3, 8 and 13: the simulations for the module I and particularly III are – in my opinion - not well corresponding to the experimental data points. Module II instead is fitting very well. Can the authors comment on this?”*

Reply: The simulations were redone and are provided in the Supplementary Figures 4 and 15. The fitting were improved. The resulting rate constants fit well with the simulated rate constants of the systems.

18. *“Also, I would suggest to improve the nomenclature of the simulated/experimental curves. In the suppl. figures 3, 8 and 13, a/a’ and b/b’ refer to a different ratio of the duplex strands in the module. In other suppl. figures and in the main manuscript, the a/a’, b/b’ and c/c’ curves refer to different conc of trigger (T1, T2 or T3). Furthermore, the modules are made with duplexes A/A’, B/B’ and C/C’ which makes all more complicated to follow.”*

Reply: We followed the request of the reviewer to change the symbols identifying the curves displayed in Supplementary Figures 4, 10 and 15. And the symbols defining the transient curves in the main text were changed into (i), (ii) and (iii) as requested.

19. *“.....it is not clear to me why the fits in panels I are done with a hyperbolic function*

and the 1st derivatives in panels II are not done on the fits. So what is the kind of function that actually better describes the experimental points? is it a hyperbolic or sigmoidal function?”

Reply: Unfortunately, we feel that reviewer missed the procedure described in the text to derive Figure 4d and f shown in Panels I, and the related derived data presented in Figure 4d and f, Panels II. In Panels I, we present in the squares the experimental fluorescence intensities corresponding to the chemical transformations shown in c and e, respectively. The purple solid line is not a simulated fitted curve (neither hyperbolic nor sigmoidal fitting). The solid line may be considered as a predictive kinetic curve presenting the temporal fluorescence changes, associated with the respective transformation, using the kinetic model (one may realize that a very good fit between the experiments and predictive calculation exist). The transient formation rates of the fluorescent products shown in Panels II correspond to the first order derivative of the experiment fluorescence changes shown in Panels I for the respective transformations. We believe that the text in the paper explains well how the respective Panels I and Panels II in Figure 4 were deduced.

20. *“the sequences of the “G genes” are not provided in the Supplementary materials.”*

Reply: The sequences of the “G genes” are now provided in the Supplementary Table 7.

Reviewer #2

We appreciate the reviewer’s comments that “the manuscript demonstrates a nice concept of cascability and interconnected DNA based non-equilibrium reaction networks”..... and that “the manuscript definitely elevates the complexity, particularly coupling of the output of three layers of cascaded reaction network to function is intriguing”

The following corrections addressing the reviewer’s comments were added to the paper:

1. *“Figure 1a authors have shown the amount of DNAzyme formed on addition of the trigger. But the curve never reaches its initial position and the deviation from the initial state increases with increase of trigger concentration. I guess these must be due to some unwanted interaction between the DNAs. Can the authors shed light on some of the side reaction taking place during the reaction network?”*

Reply: The origin for the minute incomplete recovery of the initial state is attributed to the hybridization of the waste product H_{1b} with H_{1a}. This is now explicitly explained on page 7-8.

2. *“The authors say that the recovery rate of the rest module slows down with increase of trigger concentration. I cannot see a curve in supporting information which quantitatively plots the differences of their lifetime. I suggest the authors to include a figure in SI which shows differences of their lifetime.”*

Reply: As requested by the reviewer, the half-life of the transient recovery rates at different concentrations are provided, in Supplementary Figures 6-7, 12, 22, 24, 33.

3. *“The fits reported in Supplementary Figure 3 are poor. I suggest to cut down the saturated portion and fit the data again to get the correct values.”*

Reply: The simulations were redone. The fittings were improved. The rate constants fit well with the simulated rate constants of the systems.

4. *“Can the author also include the blank where addition of hairpin H1 does not initiates layer 2?”*

Reply: The requested control experiment is provided in Supplementary Figure 21 and the results are mentioned in the text, page 11.

5. *“In figure caption figure heading are a bit confusing..... Similarly supplementary figures should also include concentration of the species used during the study for better readability.”*

Reply: The symbols defining the transient curves in the main text were changed into (i), (ii) and (iii) as requested.

6. *“In figure 2d why the second cycle shows an increased yield?..... Also, what is the difference in lifetime during second cycle?.....”*

Reply: The apparent increase in the yield of the second cycle originates from the non-zero fluorescence value upon triggering the second transient cycle. The non-zero fluorescence value originates from the slightly incomplete regeneration of the parent module due to the side hybridization of the waste products H_{1b}/H_{1a} and H_{2b}/H_{2a}, as explained in the text, page 7. The half-life for the transients shown in Supplementary Figure 22 was specified in the respective Figure captions. The half-life for the second transient of DNAzyme T₂/B was not specified due to the incomplete recovery to the initial state.

7. *“In the reaction network design, the two layers are coupled using module S/T₂. I wonder what is the advantage of using a module in between? Why the released H₂ cannot directly participate in the second layer?”*

Reply: It should be noted that the coupling unit S/T₂ is essential to communicate the two layers. Direct coupling of H_{1b} with the second layer would require complementarity between H_{1b} and B and, thus, an immediate crosstalk between B and H₁ which would perturb the communication process. The need for the coupling unit S/T₂ is also explained in the text, page 11.

8. *“For the three-layer cascaded system, the second layer reaches peak value earlier (108 minutes in Figure 2 and 35 minutes in figure 3). What is the reason? Again, is it due to different concentration?”*

Reply: The reviewer should note that for the three-layer cascade, the waste product of module II is consumed by displacing the unit H_{2b}/M. As a result, the competitive binding of H_{2b} to H_{2a} is hindered and the recovery of the module II will be enhanced. Consequently, the peak concentration of module II will appear at a time (35 minutes) that is shorter than the time (108 minutes) observed in the absence of module III. The issue of time differences for reaching the peak transient concentrations in the systems is addressed shortly on page 16 and further discussed in the supporting information, page 40.

9. *“The caption of Supplementary Figure 27 reads “The transient formation of the constituent C’ predicted by the three-layered cascade.” Predicted by what?”*

Reply: The text page 20 was corrected. The Figure caption corresponding to Supplementary Figure 34 was corrected and further explained.

10. *“Supplementary Figure 26 shows a control where absence of duplex P/W does not activate the third layer of reaction network. However, the fluorescence data shows some decrease. What could be the reason behind that?”*

Reply: The reviewer misunderstood the control experiment depicted in Supplementary Figure 32, where the duplex W/P is excluded from the three-layer cascade. Exclusion of W/P does not block the activity of the third layer, but slows down the dissipative recovery of the third layer. This control experiment shed light on the function of the duplex W/P in enhancing the recovery of the third layer. In the absence of W/P, free H_{3b} is accumulated and this strand hybridizes with H_{3a}, thus inhibiting the dissipative recovery of the third layer. So, a slow recovery of the third layer is still proceeding. The caption of Supplementary Figure 32 was slightly modified to further explain the issue.

11. *“The authors have utilized DNzyme activity to measure the transient generation of genes G1, G2 and G3. I wonder why the corresponding time dependent fluorescent changes are linear (Supplementary Figure 28, 29, 30, 31 etc)? The curves do have any saturation and how do the authors calculate the rate from that? Since it is a catalytic reaction, I am wondering they should be exponential curves rather than linear?”*

Reply: We feel that the reviewer missed the method used for the kinetic experiments validating the contents of the three Genes G₁, G₂, G₃ through the DNzyme activities of the generated genes. As shown in Supplementary Figure 36, the rates of cleavage of the substrate of DNzyme G₁ are linear and increase with the concentration of the DNzyme. These rates represent the initial rates of cleavage of the substrate (at a short time intervals 100 minutes). Under these conditions, the substrate exists at a high concentration as compared to the concentrations of G₁, and thus the rates should be linear and not parabolic. In fact, it was our purpose to measure the linear rates of cleavage in order to derive a linear calibration curve as shown in Supplementary Figure 36(b).

All the results shown in Supplementary Figures 37, 38, 39, 40 and 35 represent the cleavage rates of variable concentrations of generated G₁, G₂ and G₃ at different snapshot time-intervals of the transient gene-generation process, and all of these concentrations are evaluated under conditions where the substrates of G₁, G₂ and G₃ are at high concentrations in respect to the DNzymes. Accordingly, the rates of cleavage by the genes should be linear as reflected by the experiments.

12. *“What was the purpose of coupling it to three gene replication machineries? I guess one would have sufficient to show the concept. We do not get any extra information by coupling it to three gene replication machinery.”*

Reply: The reviewer is correct that, in principle, the coupling of the three-layer cascade to the synthesis of only one gene could be enough to demonstrate the gene-synthesis principle. This is now emphasized on page 21. We wished, however, to demonstrate that one common promoter could activate the multiple synthesis of several genes, and the three-layer cascade provides a means to control the rates of the synthesis of the different genes. These features add complexity and regulation motives into the dynamic synthesis of variable genes. These features are now further explained in the respective section, page 21.

13. *“Also, it is not so clear to me why the concentration of the gene G1, G2, and G3 changes transiently? As I can understand from the scheme the formed genes G1, G2, and G3 does not undergo any kind of degradation. So, I wonder why the concentration of the DNzyme will decrease transiently in figure 4b? Does the Figure 4b means rate*

of catalytic action of the DNAzymes?”

Reply: Unfortunately, the reviewer missed our explanation of the experiment described in Figure 4b, Panels I, II and III. The experiment was precisely described in the paper page 28. In these experiments, we take at time-intervals of operating the three-layer cascade, samples that contain **transient** concentrations of C', and these activate the gene replication machinery synthesizing G₁, G₂, G₃ exhibiting the DNAzyme activity. Thus, the activity of the DNAzyme should be a mirror image of the transient concentrations of C', as, indeed, presented in Figure 4b, Panels I, II and III. The overall experiment is aimed to demonstrate the ability of a transient network to control the transient “gene” expression efficacy.

14. *“Important: if this is not the G1, G2, G3 concentration, then please measure those concentrations.”*

Reply: These are the “real” concentrations of G₁, G₂, and G₃ at time-intervals of the operation of the three-layer cascade.

15. *“What is the extent of parasitic activation in these Polymerase/Nickase reactions and in the RCA. Especially Pol/Nic networks are prone to parasitic activation, meaning G1, G2, G3 will be produced no matter whether the C' Cascade output is there. Provide measurements”*

Reply: We performed a control experiment where the polymerization/nicking machinery is coupled to samples withdrawn from the two-layer cascade that lack the product C'. The systems do not yield any traceable gene products indicating that no parasitic products are formed. The results are presented in Supplementary Figures 49-52, and the results of these control experiments are mentioned in the text, page 21.

16. *“I would also expect the gene replication machinery to reach the peak value later than the three layers of cascade. Can the author co plot them to see that the gene replication machinery is delayed compared to the three layers of connected cascade?”*

Reply: We think that the time profile of corresponding to the transient generation of the genes should follow the time profile of the transient promoter C' (peak content after ca. 1.5 hour, cf. Supplementary Figure 34 (Predicted curve)). Using the method described in the paper to follow the transient formation of the replicated genes, it is difficult to extract a precise time where the transient concentrations of the genes reach a maximum value. Nonetheless, from the results presented in Figure 4b, Panels I-III, one may estimate that the highest transient concentration values of the genes are reached after ca. 1-1.5 hours, consistent with the transient concentration profile of C'.

17. *“Important: Almost all data appears to be single point measurements. At least duplicates need to be provided.”*

Reply: We added to the method section that all transients curves presented in the study were reproduced by $N = 3$ experiments. The transient curves demonstrated a $\leq 3\%$ deviation in the different experiments.

18. *“..... The ratio of determined values, vs fitted parameters needs to be discussed. If the ratio is completely off on the fitted values, the fits are not really meaningful as there can be so many local solutions to this problem.”*

Reply: The issue of computational modeling and fitting of the results to the experiments is now addressed in a specific discussion in the method section, page 30-31, that addresses the comment of the reviewer.

Reviewer #3

We appreciate that the reviewer found the paper “interesting paper showing that a transient three-layer DNAzyme cascade network could be fabricated and used to trigger biocatalytic processes”.

The comments of the reviewer were addressed as follows:

1. *“Similar DNAzyme-based concept has been reported many times in previous publications of the same group.....The authors should fully describe the novelty of current manuscript.”*

Reply: It should be noted that the list of references mentioned by the reviewer: JACS, 2020, **142**, 52, 21577; JACS, 2020, **142**, 51, 21460 relate to constitutional dynamic networks that **are not** dissipative or transient in their behaviors.

The references JACS, 2021, **143**, 1, 241 presents replication nicking processes leading to the evolution of DNA strands but does not include any dynamic or transient control. The references JACS, 2021, **143**, 42, 17622; JACS, 2021, **143**, 31, 12120; JACS, 2021, **143**, 13, 5071; JACS, 2020, **142**, 41, 17480 present, indeed, dynamic, transient, systems, yet they address an entirely different aspect related to dissipative transient networks: (i) They use a nicking enzyme as a catalytic unit to drive the transient systems. Namely, all of these systems require an enzyme as a catalytic guide to drive the transient processes. This is a fundamental difference as compared to the present study that applies all-DNA, DNAzyme-driven biocatalytic processes. (ii) In these systems, gated dissipative transient systems or dynamic transient reconfiguration of constitutional dynamic networks were addressed issues that were not touched in the present study.

The significance of the present study rests on the application of DNAzymes as guiding

catalysts to operate transient networks, the application of the DNazymes to operate a three-layered cascade!, and the use of the three-layer cascade as a functional framework for the synthesis of genes. These aspects are now emphasized in the conclusion paragraph of the paper, page 25.

2 *“In this manuscript, the DNzyme cascade still worked in the buffer solution. Can the authors test this DNzyme-based system in physiological environments (eg, cell lysate, living cell membranes or cytoplasm) to investigate their functions?.....”*

Reply: The suggestion of the reviewer was followed, we performed the single layer (module I), two-layer cascade (module I + II) and three-layer cascade (module I + II + III) in MCF-10A epithelial breast cell lysate, and in MDA-MB-231 breast cancer cell lysate. The results are presented in Supplementary Figures 55-57, accompanying discussion see Supplementary page 64 and mentioned in the main text, page 24-25. Experiments in cells are, however, beyond the scope of the present study.

3 *“What is the stability of the DNzyme cascade network in the working solution and in the physiological environments?.....”*

Reply: The DNA networks reveal stabilities in the cell lysates for at least three days. A comment recording the stability of the network was added to the text, page 25.

4 *“All the DNzyme networks used in the paper are Mg²⁺-ion-dependent. Is the concentration of Mg²⁺ ions in the physiological environment sufficient to drive the DNzyme cascade to perform the corresponding functions?”*

Reply: The concentrations of Mg²⁺ in cells correspond to 0.5-1.5 mM. References to these values are provided in the supplementary material, page 77. These concentrations are sufficient to operate the networks and in vitro, pure buffer conditions or in cell lysates. Results are provided on page 68, supplementary information. The conclusion part was modified to address this comment of the reviewer.

REVIEWER COMMENTS

Reviewer #1 (Remarks to the Author):

2. ".....I am not sure to see the advantage of using a transient process to perform the target reactions. The transient dissipative nature of the system is in my opinion not really exploited, rather observed, and instead the waste is used to trigger another cycle."

"Maybe to prove or disprove this concept, the authors should show what happens if the polymerase reaction depicted in figure 4a is fueled by a defined concentration of C', however, in absence of the entire cascade. This may better evidence the impact of the cascade and the advantage of having it....."

Reply: We feel that reviewer missed the functions of the transient DNAzyme layer in controlling the modulated transient formation of the "genes" G1, G2, G3. In fact, his/her suggestion to probe the transient features of C' on the emerging transient features of the genes G1, G2, G3 by applying a constant concentration of C' on the emergence of the genes is excellent. This experiment was performed and is described on page 20 and the results are shown in the Supplementary Figures 41-44. Under these conditions, the formation of the genes and their catalytic functions stay constant, and thus, the transient cascaded network, indeed, dictates the emerging dissipative properties of the DNAzyme genes. This issue is further emphasized in an explicit explanation on page 21.

Comment reviewer#1: I appreciate that the authors followed my suggestion. Still I believe that there is an over interpretation of the role of the energy dissipation in this experiment. Also, the G1 to G3 are repeatedly called "genes". The authors maybe refer to the fact that they "code" for DNAzymes, which have catalytic properties. If true, this should be more clearly stated in the text.

"It would be also helpful to state if the substrate molecules S1, S2 and S3 are already present in solution and at which concentration. To my understanding, in presence of magnesium ions and excess substrate molecules, G1, G2 and G3 sequences form transient DNAzymes, according to a mechanism that is very similar to the one displayed by the DNAzymes of the cyclic reactions. If this is true, then it would not be surprising to observe a transient behavior of activity, even in absence of the cascade."

Reply: We feel that the reviewer missed the way to derive the temporal concentrations of the genes generated by the transient three-layer cascade, displayed in Figure 4b. In fact, the procedure was detailed in the original manuscript page 28 and the information requested by the reviewer was presented in that section.

For the sake of clarity, we repeat step-by-step the procedure to evaluate these concentrations:

- (i) We take samples out of the three-layer cascade at different time-intervals of operation.
- (ii) We apply the replication/nicking machinery synthesizing the genes G1, G2, G3 on each of the samples for a time-interval of six hours.
- (iii) After completion of the synthesis of the genes, the substrates S1, S2, S3 are added at an excess (2 μ M), and the concentrations of the synthesized genes are quantitatively evaluated by following the rates of cleavage of the substrates for a short time interval (100 minutes). The derived concentrations represent the temporal behavior of the formation of the respective genes.

Comment reviewer#1: The procedure is now clearer, despite I find this method of concentration determination quite complicated. If I understand correctly, the concentrations (y-axis) of fig. 4b refer to the concentration of the G species at different time intervals of the three-layered cascades. Each point is actually measured during the initial life-time (100 min) of the transient species, formed by the G/S DNAzyme/substrate complex (this method has been largely used by the authors in previous papers). The authors should more clearly state this point in the text.

Minor points:

4. "More details about the simulation procedure should be given, such as the initial conditions of the system and the law of mass conservation."

Reply: As requested, further details on the simulation procedures are provided in the "Methods" section, page 30, and the initial conditions of the system were added to the text. The simulation procedure followed to the law of mass conservation, and this is explicitly stated in the section describing the simulation process.

Comment reviewer#1: The authors should clearly state if the simulations presented in the concentration vs time curves are referred to $dT1A/dt$ or $dT1AH1/dt$ (for module I). Both species contain the strand A which is the source of the Cy5 fluorescence signal measured over time. Indeed, both species are transient and both species will lead to a decrease of Cy5 fluorescence signal over the course of the cyclic reaction.

The same comments apply to the module II and module III.

Two control experiments should be added:

(1) kinetics of module II in absence of module I (BB', H2 and Mg ions in presence of ST2 only) to evaluate the spurious hybridization of B to T2.

(2) similarly, kinetics of module III in absence of module II to evaluate the spurious hybridization of T3 to C.

Fig. 3: panels I, II and II in b and c are referred to the isolated modules? This is not stated in the legend and creates confusion with the kinetic profiles in panel IV.

In general, ALL graphs reporting concentration vs time should clearly state in the label to the y-axis to which species the concentration is referred to.

8. "please also state the half-life of the DNAzyme decay in each cycle."

Reply: The half-life of the DNAzyme decay transients were added in the Supplementary Figures 6-7, 12, 22, 24, 33.

Comment reviewer#1: Half-life vs cycle (suppl. Fig. 6b and 7e and similar) should be presented differently. in this way, it looks like an interpolation between two points, which has obviously no scientific meaning.

15. "... I believe it would be extremely useful to add here a non-denaturing gel characterization of the species present in solution at different times. This should help to visualize transient species which are not fluorescently labelled and might be also of help to validate the rate coefficients simulated. Indeed, the Matlab code can be easily modified to generate the time profile of any desired species of the ODE system. Such theoretical predictions can be easily compared with the appearance/disappearance and relative intensity of the gel bands associated to the distinct intermediate species. This would strongly support the theoretical model."

Reply: Although the experiment suggested by the reviewer to support the transient behavior of the systems was a challenge, we appreciate this constructive suggestion of the reviewer. This experiment was performed and is addressed in the text, page 7, and presented in the supporting information, page 4. The transient electrophoretically separated bands fit well with the predicted transient concentrations, using the kinetic model.

Comment reviewer#1: I appreciate the efforts of the authors. However, the gel should be a bit more described. What are the bands appearing from lane 2 to 8, which migrate slower than the T1/H1a species? why do they increase in intensity and in molecular weight over time? This is also accompanied by a similar trend in the low MW bands which migrate around the AA' and H1

species. Also, the quality of the fit is in my opinion not "very good" as stated by the authors.

Reviewer #2 (Remarks to the Author):

Revisions are well done and thorough. The paper can be accepted.

Reviewer #1

Comment #1: *“I appreciate that the authors followed my suggestion. Still I believe that there is an over interpretation of the role of the energy dissipation in this experiment. Also, the G1 to G3 are repeatedly called “genes”. The authors maybe refer to the fact that they “code” for DNAzymes, which have catalytic properties. If true, this should be more clearly stated in the text.”*

Reply: We followed the suggestion of the reviewer to alter the term “genes” for G₁ to G₃ into the term “coded-strands” and changed the specific paragraph, conclusion section and title accordingly. We feel, however, that the dynamic synthesis of these different “coded strands” could provide a model for gene replication and this was mentioned in the respective paragraph and title.

Comment #2: *“The procedure is now clearer, despite I find this method of concentration determination quite complicated. If I understand correctly, the concentrations (y-axis) of fig. 4b refer to the concentration of the G species at different time intervals of the three-layered cascades. Each point is actually measured during the initial life-time (100 min) of the transient species, formed by the G/S DNAzyme/substrate complex (this method has been largely used by the authors in previous papers). The authors should more clearly state this point in the text.”*

Reply: As requested, the method to evaluate the concentrations of the coded DNAzyme strands was further detailed and explained in the Methods section, page 29 and additional details explaining the method were introduced into the supplementary material, page 47. This page outlines the stepwise procedure to evaluate the concentrations of the DNAzymes.

Minor points

Comment #3: *“The authors should clearly state if the simulations presented in the concentration vs time curves are referred to $dT1A/dt$ or $dT1AH1/dt$ (for module I). Both species contain the strand A which is the source of the Cy5 fluorescence signal measured over time. Indeed, both species are transient and both species will lead to a decrease of Cy5 fluorescence signal over the course of the cyclic reaction.”*
“The same comments apply to the module II and module III.”

Reply: As suggested by the reviewer, we explicitly stated on page 31 to page 32 that the concentrations vs. time curves depicted for module I, module II and module III correspond to the DNAzyme T₁/A (including the free DNAzyme T₁/A and the T₁/A transient binding with H₁ and the cleaved H₁), in module I, DNAzyme T₂/B (including the free DNAzyme T₂/B and the T₂/B transient binding with H₂ and the cleaved H₂), in module II, and DNAzyme T₃/C (including the free DNAzyme T₃/C and the T₃/C

transient binding with H₃ and the cleaved H₃), in module III. We also explained that for the simulation of the transient curves we derived the concentration changes of A' (dA'/dt), B' (dB'/dt) and C' (dC'/dt) that equal to the fluorescence time-dependent changes corresponding to dA/dt, dB/dt and dC/dt, respectively.

Comment #4: *“Two control experiments should be added: (1) kinetics of module II in absence of module I (BB', H2 and Mg ions in presence of ST2 only) to evaluate the spurious hybridization of B to T2. (2) similarly, kinetics of module III in absence of module II to evaluate the spurious hybridization of T3 to C.”*

Reply: The control experiments were performed and the results are presented with Supplementary Figure 20 and Supplementary Figure 30, and discussed shortly in the text, pages 11-12 and page 15.

Comment #5: *“Fig. 3: panels I, II and III in b and c are referred to the isolated modules? This is not stated in the legend and creates confusion with the kinetic profiles in panel IV.”*

Reply: Indeed, as suggested by the reviewer, we added to the figure caption of Figure 3 the statements that the kinetic curves presented in Panel I correspond to DNAzyme T₁/A of module I, Panel II correspond to DNAzyme T₂/B of module II, and Panel III correspond to DNAzyme T₃/C of module III.

Comment #6: *“In general, ALL graphs reporting concentration vs time should clearly state in the label to the y-axis to which species the concentration is referred to.”*

Reply: We rechecked all figure captions and they include explicit identification of the species associated with the y-axis transient constituents of the respective systems.

Comment #7: *“Half-life vs cycle (suppl. Fig. 6b and 7e and similar) should be presented differently. in this way, it looks like an interpolation between two points, which has obviously no scientific meaning.”*

Reply: The reviewer is correct. Supplementary Figures 6b, 7e, 12e, 23b and 25b were corrected, as requested.

Comment #8: *“I appreciate the efforts of the authors. However, the gel should be a bit more described. What are the bands appearing from lane 2 to 8, which migrate slower than the T1/H1a species? why do they increase in intensity and in molecular weight*

over time? This is also accompanied by a similar trend in the low MW bands which migrate around the AA' and H1 species. Also, the quality of the fit is in my opinion not "very good" as stated by the authors."

Reply: The gel electrophoretic results provided in Supplementary Figure 3 were further discussed and explained on page 4 in the supplementary material, as requested.